# Learning-Augmented Algorithms with Explicit Predictors

**Marek Eliáš**
Bocconi University, Milan

**Haim Kaplan**
Tel Aviv University
Google Research

**Yishay Mansour**
Tel Aviv University
Google Research

**Shay Moran**
Department of Mathematics, Technion
Department of Computer Science, Technion
Department of Data and Decision Sciences, Technion
Google Research.

## Abstract

Recent advances in algorithmic design show how to utilize predictions obtained by machine learning models from past and present data. These approaches have demonstrated an enhancement in performance when the predictions are accurate, while also ensuring robustness by providing worst-case guarantees when predictions fail. In this paper we focus on online problems; prior research in this context was focused on a paradigm where the algorithms are oblivious of the predictors' design, treating them as a black box. In contrast, in this work, we unpack the predictor and integrate the learning problem it gives rise for within the algorithmic challenge. In particular we allow the predictor to learn as it receives larger parts of the input, with the ultimate goal of designing online learning algorithms specifically tailored for the algorithmic task at hand. Adopting this perspective, we focus on a number of fundamental problems, including caching and scheduling, which have been well-studied in the black-box setting. For each of the problems, we introduce new algorithms that take advantage of explicit and carefully designed learning rules. These pairings of online algorithms with corresponding learning rules yields improvements in the overall performance in comparison with previous work.

## 1 Introduction

We study online algorithmic problems within the realm of learning-augmented algorithms. A learning-augmented algorithm possesses the capability to work in conjunction with an oracle that supplies predictive information regarding the data it is expected to process. This innovative approach has been discussed in landmark studies by Kraska et al. [2018] and Lykouris and Vassilvitskii [2021], situating it neatly within the "beyond worst-case analysis" framework [Roughgarden, 2020, chap. 30].

In this framework, studies typically define predictions specifically tailored to the problem at hand, which could presumably be learnt from historical data. Predictions might include, for instance, the anticipated next request time for a page in a caching problem or the expected duration of a job in a scheduling task. These predictions are accessible either before or together with the online requests, allowing the algorithm to utilize them for performance enhancement (measured by competitive ratio or regret). The objective is for the algorithm's performance to gracefully decrease as prediction accuracy declines, ensuring it never underperforms the baseline achievable without predictions.

38th Conference on Neural Information Processing Systems (NeurIPS 2024).

Despite the elegance of these results, several important aspects were neglected: the ad-hoc nature of the predictions, the lack of standardized quality measures, and the frequently overlooked prediction generation methods and their relation to the algorithmic process. We believe that addressing these aspects is likely to yield substantial improvements.

Consider week-day and festive traffic patterns in a city – a simple example of a setting with two typical inputs requiring very different predictive and algorithmic strategies. Achieving a good performance in such setting requires a learning component in the algorithm which discerns between the festive and week-day input instances and suggests an appropriate routing strategy. Such learning components are already present (explicitly or implicitly) in works on combining algorithms in a black-box manner [Dinitz et al., 2022, Emek et al., 2021, Anand et al., 2022, Antoniadis et al., 2023], where a switch between algorithms is made after incurring a high cost.

Our approach goes one step further. It is based on making the computation of the predictions an integral part of the algorithmic task at hand. We do this by making all the data (historical and current) directly available to the online algorithm (rather than summarizing it into ad-hoc predictions). This gives the algorithm two important abilities. The first ability is to learn the input instance based on its prefix (the shorter the better) and adapt the algorithmic strategy before incurring significant cost. E.g., in the example above, week-day and festive traffic patterns can be easily discerned already in early morning when the traffic is low and possibly suboptimal routing decisions have negligible impact on the overall cost. The second ability is to identify actions which are beneficial for many plausible input instances simultaneously but are not suggested by any of the black-box predictors. We use both abilities in design of our algorithms.

In more detail, we model the past data through the assumption that the algorithm is equipped with prior knowledge comprising a set of descriptions of past input instances. Each description offers specific statistics or characteristics that represent essential information of the input instance. Borrowing terminology from learning theory, we call this set a *hypothesis class* and denote it by $\mathcal{H}$. More specifically, $\mathcal{H}$ is a set of hypotheses, where each hypothesis, $h(I)$, consists of information regarding a specific possible input instance $I$ of the algorithmic task.

In the simplest setting each hypothesis could be the input instance itself ($h(I) = I$). Like the sequence of pages to arrive in a caching instance, or a set of jobs to arrive in a scheduling instance. In other situations, an hypothesis $h(I)$ could be a more compact summary of the instance $I$, such as the distribution of the arriving jobs (what fraction are of each "type"). In such case, many past input instances will correspond to the same hypothesis and the size of $\mathcal{H}$ will be much smaller than the size of the dataset of past instances. However, in all cases that we consider, each hypothesis $h(I)$ provides sufficient information about the instance in order to determine an offline optimal solution $\mathrm{OPT}(I)$.

We distinguish between *realizable* and *agnostic* settings. In the *realizable* setting, we make the assumption that the actual input instance that the online algorithm has to serve, perfectly aligns with one of the hypotheses in $\mathcal{H}$. That is, if $I$ is the real input, then $h(I) \in \mathcal{H}$. In the *agnostic* setting we remove this assumption and consider arbitrary inputs. Our goal is to deliver performance guarantees that improve if the actual input is "close" (defined in a suitable manner) to the hypothesis class $\mathcal{H}$. Agnostic setting with $|\mathcal{H}| = 1$ captures the (usual) setting of ML-augmented algorithms with a single black-box predictor. The realizable case is interesting mostly from a theoretical perspective as a very special case of the agnostic setting. Its simplicity makes it a logical starting point of a study.[1] If the current instance does not match any hypothesis perfectly (in the realizable setting) or is far from $\mathcal{H}$ (in the agnostic setting), we can still achieve good performance using robustification techniques, see e.g. [Wei, 2020, Antoniadis et al., 2023, Lattanzi et al., 2020, Lindermayr and Megow, 2022].

Our methodology is to split the algorithm into two parts. One (called *predictor*) produces predictions based on the provided hypothesis class ($\mathcal{H}$) and the part of the input seen so far. Its goal is to produce the best prediction for the instance at hand which could be a hypothesis from $\mathcal{H}$ or some other suitable instance designed based on $\mathcal{H}$. The second part is the online algorithm itself. It uses the prediction of the first part to serve the input instance with low cost. In particular, it can compute the offline optimal solution for the prediction and serve the input instance according to this solution.

The predictor is the learning component of our algorithm. It solves a learning task which is associated with the algorithmic problem at hand. For example, the learning task associated with caching is

---

[1]Boosting technique, which had a great impact on applied and practical machine learning, was developed while studying relationship between weak and strong PAC learning in the realizable setting.

| | caching | load balancing | non-clairvoyant scheduling |
|---|---|---|---|
| realizable | OPT$+k\log\ell$ | $O(\log\ell)\,$OPT | OPT$+\ell\sqrt{2\,\text{OPT}}$ |
| agnostic | OPT$+O(\mu^* + k\log\ell)$ | $O(\log\ell)\,$ALG$^*$ | OPT$+\mu^* + O(n^{5/3}\log\ell)$ |
| previous works | OPT$+O(\mu^* + k + \sqrt{Tk\log\ell})$ | $O(\log\ell)\,$ALG$^*$ | $(1+\epsilon)\,$OPT $+O(1/\epsilon^5)\mu^*$ |
| | [Emek et al., 2021] | [Dinitz et al., 2022] | [Dinitz et al., 2022] |

Figure 1: Summary of our results. Notation: $\ell = |\mathcal{H}|$; $k$ and $T$: cache size and instance length respectively in caching; $m$: the number of machines in load balancing; $n$: the number of jobs in non-clairvoyant scheduling; $\mu^*$: distance of the input from the hypothesis class in caching and non-clairvoyant scheduling; ALG$^*$: cost of the best algorithmic strategy suggested by $\mathcal{H}$.

a variant of online learning with two kinds of costs: the smaller cost is due to a misprediction of an individual request and the larger one due to *switching* to a different predicted sequence. The costs are chosen to reflect the impact of the two events on the overall performance of the algorithm.

We consider this new way to model a setting of "online algorithm with predictions" as one of our core contributions (in addition to the algorithms for the specific problems that we describe below). In a sense, our technique interpolates in an interesting way between the learning challenge (from historical data) and the algorithmic challenge, while addressing both of them.

## 1.1 Performance bounds of our algorithms

We propose algorithms (within our framework) for three fundamental online algorithmic problems: caching, load balancing, and non-clairvoyant scheduling. For caching and non-clairvoyant scheduling, we achieve a (small) additive regret compared to the offline optimal solution instead of a multiplicative competitive ratio. For load balancing, we achieve a competitive ratio with logarithmic dependence on the size $\ell$ of the hypothesis class. Our results are summarized in Figure 1, while the full description is deferred to Section 2.

We focus our presentation on the basic framework with performance bounds dependent on the size of the hypothesis class $\mathcal{H}$. We assume $\mathcal{H}$ to be restricted in the sense that not every instance $I$ is close to some hypothesis in $\mathcal{H}$. This ensures that there is some structure in the input instances which can be learnt. With an unrestricted $\mathcal{H}$, every input would be possible and we would be in the classical online setting. However, our modular approach allows replacing the learning component in order to achieve additional desirable properties. This includes fast running time, better performance with clusterable datasets (see Dinitz et al. [2022]), and better performance on instances composed of several parts, each resembling a different hypothesis (see [Anand et al., 2022, Antoniadis et al., 2023]).

Recent works by Dinitz et al. [2022] and Emek et al. [2021] consider algorithms with access to a portfolio of predictors trying to achieve performance comparable to the best one. Our results can be interpreted in their setting by considering the output of each predictor in the portfolio as a hypothesis. We achieve comparable and sometimes better results (see Figure 1 for comparison) using an arguably simpler approach, separating the learning and algorithmic part and solving them separately.

**Organization.** Section 2 describes our main contributions including the description of the problems studied and the approach which leads to our results. The survey of the related literature in Section 3 is followed by a warm-up in Section 4 containing an exposition of our approach on caching in realizable setting. The main technical part of our paper is in Appendix. Appendix B considers the load balancing problem and Appendix C the non-clairvoyant scheduling problem. We conclude our paper with treatment of caching in agnostic setting in Appendix D.

## 2 Main Results

Our study focuses on three fundamental online algorithmic problems: caching, load balancing, and non-clairvoyant scheduling. For each of these problems, we define learning tasks and devise explicit and efficient predictors for solving them. We demonstrate how these predictors can be integrated into algorithms designed to tackle the respective online problems. A key feature of our approach is the

modular separation of the learning and algorithmic components. By decoupling these aspects, we develop simpler algorithms that often yield improved bounds compared to previous works in the field.

## 2.1 Caching

In the caching problem, the input is a sequence of page requests. The online algorithm holds a cache of size $k$, and it must ensure that the currently requested page is always available in the cache. If the requested page is absent from the cache, a page fault occurs, prompting the page to be loaded into the cache. If the cache is already full, another page must be evicted to make room. The ultimate objective is to minimize the number of page faults.

In the offline scenario, where the input sequence is known ahead of time, an optimal algorithm adheres to the intuitive policy of removing a page that will not be requested again for the longest time. This algorithm, known as Furthest in the Future (FitF) [Belady, 1966], achieves the minimum possible number of page faults.

### The Learning Task: "Sequence Prediction with Switching Cost"

In this context we consider a variant of the classical learning task of sequence prediction that includes a switching cost. More precisely, the objective of the predictor is to predict the sequence of page requests denoted by $r_1, r_2, ..., r_T$. In each round $t$, the predictor presents a prediction for all remaining requests in the sequence $\pi_t, \pi_{t+1}, ..., \pi_T$. At the conclusion of the round, the predictor sees $r_t$ and incurs a loss of $1[\pi_t \neq r_t]$ if the prediction was incorrect. After observing $r_t$, the predictor can choose to alter the subsequent predictions to $\pi'_{t+1}, ..., \pi'_T$. Each time the predictor decides to modify the predictions, a switching cost of $k$ is incurred (remember that $k \geq 1$ represents the size of the cache). Thus, the total loss of the predictor is equal to the number of prediction errors plus $k$ times the number of switches.

**Hypotheses.** Each hypothesis in our class $\mathcal{H}$ is a possible input sequence. In the realizable scenario, we operate under the assumption that the actual input matches one of the hypotheses within the class. In the agnostic case we relax this assumption and provide guarantees that scale with the Hamming distance between the input sequence and the hypothesis class.

In the realizable case we design a deterministic predictor whose total loss is at most $k \log \ell$ (recall that $\ell = |\mathcal{H}|$). It is based on majority vote or the halving algorithm [Littlestone, 1987]. An interesting and subtle point is that our predictor is improper, meaning it occasionally predicts the remaining sequence of page requests in a manner that does not align with any of the hypotheses in $\mathcal{H}$. To incorporate such improper predictors, we need to use an optimal offline caching algorithm that is monotone in the following sense: applying the algorithm on an input sequence $r_1, \ldots, r_T$ produces a caching policy which is simultaneously optimal for all prefixes $r_1, \ldots, r_t$ for $t \leq T$. Fortunately, Belady's FitF algorithm has this property, as outlined in Observation 4.

For the agnostic setting, we design a randomized predictor with a maximum total loss of $O(\mu^\star + k \ln \ell)$, where $\mu^\star$ is the Hamming distance of the actual input sequence from the class $\mathcal{H}$. This predictor utilizes a multiplicative-weight rule [Littlestone and Warmuth, 1994], and its learning rate is specifically adapted to achieve the optimal balance between the cost of changing predictions (switching costs) and the inaccuracies in the predictions themselves (prediction errors).

Our final caching algorithm incorporates a predictor for this problem in such a way that at each round $t$, it applies Belady's FitF algorithm to the predicted suffix of the sequence $\pi_t, ..., \pi_T$. We then show that the cumulative loss of the predictor serves as an upper bound on the additional number of page faults that our algorithm experiences compared to the offline optimal algorithm. Overall we obtain the following guarantees for our caching strategy:

**Theorem 1** (Caching). *Let $\mathcal{H}$ be a hypothesis class of size $\ell$ and $I$ be an input instance with offline optimum value $\mathrm{OPT}(I)$. There is a deterministic algorithm for the realizable setting (i.e., $I \in \mathcal{H}$) which has cost at most $\mathrm{OPT}(I) + k \log \ell$. There is a randomized algorithm for the agnostic setting with expected cost at most $\mathrm{OPT}(I) + (5 + 6/k)\mu^\star + (2k + 1) \ln \ell$, where $\mu^\star$ is the Hamming distance between $I$ and the best hypothesis in $\mathcal{H}$.*

Our algorithms can be robustified, i.e., we can ensure that their cost is not larger than $O(\log k)\,\mathrm{OPT}(I)$ while loosing only a constant factor in the dependency on $\mu^\star$ and $\ln \ell$ in our

additive regret bound, see Section D.1 for details. Note that the previous methods used to achieve robustness for caching usually lose an additive term linear in $\mathrm{OPT}(I)$, see [Wei, 2020, Blum and Burch, 2000, Antoniadis et al., 2023]. In Section D.2, we describe how to extend our results to the setting where each hypothesis, instead of a complete instance, is a set of parameters of some prediction model producing next-arrival predictions. In Section D.3, we show that the dependency on $\ell, k$, and $\mu^\star$ in Theorem 1 cannot be improved by more than a constant factor. Our result is an improvement over the $O(\mu^\star + k + \sqrt{Tk \log \ell})$ regret bound of Emek et al. [2021] whenever $T = \omega(k \log \ell)$.

## 2.2 Load Balancing

In online load balancing on unrelated machines, we have $m$ machines numbered from 1 to $m$ and a total of $n$ jobs. The jobs arrive sequentially, and the objective is to assign each job to one of the machines upon its arrival in order to minimize the *makespan*, which is the total time that the busiest machine is actively working. Each job is characterized by its type, which is an $m$-dimensional vector $p$. The value $p(i)$ indicates the time required for the $i$-th machine to complete the job. As the jobs arrive, the algorithm observes the job's type and makes a decision on which of the $m$ machines to schedule it. These scheduling decisions are made in an online manner, without knowledge of future jobs.

In the offline setting, the ordering of the jobs in the input sequence does not play any role. In fact, an instance of load balancing is sufficiently described by the number of jobs of each type which need to be scheduled and these numbers are available to the algorithm in advance. A 2-approximation algorithm by Lenstra, Shmoys, and Tardos [1990] based on linear programming achieves a makespan that is at most twice the makespan of the optimal schedule.

### The Learning Task: Forecasting Demand

The learning problem that arises in this context of makespan minimization is a rather natural one and might be interesting in other contexts. The goal of the predictor is to forecast, for each possible job type $p$, the number of jobs of type $p$ that are expected to arrive. The predictor maintains a prediction that includes an upper bound, denoted as $n_p$, on the number of jobs of each possible job type $p$. Similar to caching, the learning problem involves two distinct costs: prediction errors and switching costs. A prediction error occurs when the actual number of jobs of a particular type exceeds the predicted upper bound $n_p$. The cost of a prediction error is determined by the type of the job that witnessed it. A switching cost occurs when the predictor decides to modify its prediction (i.e., the predicted upper bounds $n_p$'s). The cost of such a modification is the makespan associated with the new prediction.[2]

**Hypotheses.** In load balancing each hypothesis $f$ in our class $\mathcal{H}$ predicts the frequency of the jobs of each type $p$. That is, for each type $p$ it assigns a number $f_p \in [0, 1]$ which represents the fraction of jobs in the input whose type is $p$. We stress that the hypothesis does not predict the actual number of jobs of each type, nor does it even predict the total number of jobs in the input. In practice, the numbers $f_p$ can be estimated by past statistics. With the knowledge of the correct hypothesis $f$, we are able to produce an integral assignment of jobs to machines at a low cost. Previously studied machine-weight predictions [Lattanzi et al., 2020] allow producing a fractional assignment which requires a costly rounding procedure [Li and Xian, 2021].

In the realizable scenario, we operate under the assumption that the actual input matches one of the hypotheses within the class. In the agnostic case we relax this assumption and provide guarantees that scale with the maximum (multiplicative) approximation error between the true frequencies and those predicted by the best hypothesis (see below).

In the realizable case, we design a simple randomized predictor, ensuring that the total expected loss is at most $O(\mathrm{OPT}(I) \cdot \log \ell)$ (recall that $\ell = |\mathcal{H}|$), where $\mathrm{OPT}(I)$ represents the makespan of the input instance $I$.[3] The key idea is to guess the total number of jobs in the input sequence and accordingly to scale the frequencies in each hypothesis to predict the number of jobs $n_p$ of each

---

[2]Note that the offline optimal makespan does not depend on the order of the jobs, it only depends on the number of jobs of each type, and hence, it is a function of the predicted numbers $n_p$ for the types $p$.

[3]We refer to the makespan of the optimal schedule for an instance as the makespan of the instance.

type. The randomized predictor maintains a random hypothesis consistent with the processed jobs. Whenever one of the predicted counts $n_p$ is violated, the predictor switches to a randomly chosen consistent hypothesis from $\mathcal{H}$, resembling the classical randomized marking strategy in caching [Fiat et al., 1991].

We additionally present a deterministic predictor with loss of at most $O(\mathrm{OPT}(I) \cdot \log(|\mathcal{H}|) \cdot \log \tau)$, where $\tau$ is the number of job types with non-zero frequency in at least one of the hypotheses. The deterministic rule predicts the median among the counts $n_p$ provided by the hypotheses for each job type $p$. The analysis of this deterministic learning rule is more intricate than that of the randomized one. The crucial insight is that the produced "medians" prediction can be scheduled within makespan at most $O(\mathrm{OPT}(I) \log \tau)$. Our predictors in the agnostic setting are based on those in the realizable case.

Our scheduling algorithm incorporates a predictor for this problem in such a way that at each round $t$, it behaves in accordance with the algorithm of Lenstra et al. [1990], applied to the predicted upper bounds $n_p$'s. We demonstrate that the cumulative loss of the predictor serves as an upper bound on the makespan. We obtain the following result:

**Theorem 2** (Load balancing). *There are algorithms using a deterministic and randomized predictor respectively which, given a hypothesis class $\mathcal{H}$ of size $\ell$ and an instance $I$ with makespan $\mathrm{OPT}(I)$, satisfy the following. In the realizable setting (i.e., $h(I) \in \mathcal{H}$, where $h(I)$ is the distribution corresponding to $I$), they produce a schedule whose makespan is at most $O(\log \ell \log \tau \, \mathrm{OPT}(I))$ and $O(\log \ell \, \mathrm{OPT}(I))$ in expectation, respectively, where $\tau$ is the number of job types with non-zero frequency in at least one of the hypotheses. In the agnostic case they produce a schedule with makespan at most $O(\alpha\beta \log \ell \log \tau \, \mathrm{OPT}(I))$ and $O(\alpha\beta \log \ell \, \mathrm{OPT}(I))$ in expectation, respectively, where $\alpha$ and $\beta$ describe the multiplicative error of the best hypothesis $f^\star \in \mathcal{H}$.*

In agnostic case, the multiplicative error of hypothesis $f$ with respect to an instance with frequencies $f^*$ is defined as follows. If there is a job type $p$ such that $f_p \neq 0$ and $f_p^\star = 0$, we define $\alpha := n + 1$, where $n$ denotes the number of jobs in the input instance. Otherwise, we define $\alpha := \max\{f_p/f_p^\star \mid f_p^\star \neq 0\}$. Similarly, if there is a job type $p$ such that $f_p = 0$ and $f_p^\star \neq 0$, we define $\beta := n + 1$. Otherwise, $\beta := \max\{f_p^\star/f_p \mid f_p \neq 0\}$. We have $\alpha, \beta \leq n + 1$.

Our algorithms can be robustified so that their competitive ratio[4] is never larger than $O(\log m)$ (the best possible competitive ratio in the worst-case setting [Azar et al., 1992]), while loosing only a constant factor in the bounds mentioned in Theorem 2, see Section B.5. In Section B.7, we show that our competitive ratio in the realizable case cannot be improved by more than a constant factor.

Previous works focused on predictions assigning weight to each machine which indicates its expected load [Lattanzi et al., 2020], and acquiring a solution for the fractional variant of the problem. Dinitz et al. [2022] showed how to aggregate outputs of $\ell$ algorithms into a single fractional solution, loosing a factor of $O(\log \ell)$ compared to the best of the algorithms. A fractional solution can be rounded online, loosing a factor of $\Theta\left(\frac{\log \log m}{\log \log \log m}\right)$ in the competitive ratio [Lattanzi et al., 2020, Li and Xian, 2021]. Instead, we use job-type frequencies which allow us to produce an integral solution directly without the costly rounding procedure. However, our approach can be used to aggregate outputs of any $\ell$ algorithms, preserving integrality of their solutions, see Section B.6.

## 2.3 Non-clairvoyant Scheduling

We consider a non-clairvoyant scheduling problem in which a single machine is assigned the task of completing a set of $n$ jobs, denoted as $j_1, j_2, \ldots, j_n$. The jobs are released at time $0$ and the scheduler's objective is to determine the order in which they should be scheduled such that the sum of their completion times is minimized.

The optimal ordering is obtained by sorting the jobs in ascending order of their processing times. However, in the non-clairvoyant setting, the scheduler does not know these processing times. To address this challenge, the scheduler is allowed to preempt a job before it is completed, meaning that it can interrupt the ongoing execution of a job and replace it with another job. The remaining portion of the preempted job is then rescheduled for completion at a later time. Round-Robin algorithm is

---

[4]The maximum ratio between the cost of the algorithm and the offline optimal solution over all instances.

2-competitive and this is the best competitive ratio achievable in non-clairvoyant setting in the worst case [Motwani et al., 1993].

**The Learning Task: Comparing Job Durations**

In the learning task explored within this context, the objective is for the predictor to learn the optimal ordering of jobs. We investigate two variants of this learning problem, one suited to the realizable setting and one suited to the agnostic setting.

In the realizable case, we adopt a similar approach to the previous sections. Here, each hypothesis within the class provides predictions for the processing times of all $n$ jobs. We then design a predictor that learns the correct hypothesis in an online fashion. Our overall scheduling algorithm in the realizable case operates by always scheduling first the job with the shortest predicted processing time.

In the agnostic setting we follow a different methodology which is more in line with statistical learning. We use here a weaker type of hypotheses: each hypothesis is a permutation of the $n$ jobs, indicating a prediction of the optimal ordering, without specifying the exact processing times.

In this learning task, the predictor is provided with a training set consisting of a small subset of the jobs that is sampled uniformly. For each job in the training set the predictor sees their lengths. Using this training set, the predictor generates a permutation $\pi$ of the $n$ jobs.

Each permutation $\pi$ is associated with a loss[5] which reflects the performance of a scheduler that follows the order suggested by $\pi$. In particular, the loss is defined in such a way that the optimal permutation has the best (lowest) loss, and more generally permutations with faster completion times have smaller losses. The predictor we design for this task uses the training set to approximate the loss of every permutation in the class $\mathcal{H}$, and outputs the one which minimizes the (approximated) loss.

In order to avoid scaling issues, we formulate our guarantees for instances with maximum job length at most 1.[6]

**Theorem 3** (Completion Time). *Consider an input instance $I$, let $\mathrm{OPT}(I)$ denote the offline optimal value of its total completion time objective, and let $\mathcal{H}$ be a hypothesis class of size $\ell$. We assume, without loss of generality, that the maximum job length in $I$ is at most 1. Then, there is a deterministic algorithm which achieves total completion time at most $\mathrm{OPT}(I) + \ell\sqrt{2\,\mathrm{OPT}(I)}$ in the realizable setting (i.e., $I \in \mathcal{H}$). In the agnostic setting, there is a randomized algorithm that with high probability achieves total completion time of at most $\mathrm{OPT}(I) + \mu^* + O(n^{5/3}\log^{1/3}\ell)$, where $\mu^*$ is the difference between the total completion time of the best hypothesis in the class and $\mathrm{OPT}$.*

Note that the value of $\mathrm{OPT}(I)$ is quadratic in $n$ unless the size of a vast majority of jobs in $I$ is either 0 or vanishing as $n$ grows.

We have also found an unexpected separation: there is an algorithm for the realizable setting with regret at most $n\log\ell$ on input instances containing jobs of only two distinct lengths (Section C.1). On the other hand, there is no algorithm with regret $o(\ell n)$ on instances with at least three distinct lengths (Section C.3).

Previous work by Dinitz et al. [2022] showed the following. For any $\epsilon > 0$, there is an algorithm which achieves expected total completion time $(1+\epsilon)\,\mathrm{OPT} + O(1/\epsilon^5)\mu^*$ under certain assumptions about the input. Therefore, their bound always gives a regret linear in $\mathrm{OPT}$ and a higher dependency on $\mu^*$.

Any algorithms can be robustified by running it at speed $(1-\delta)$ simultaneously with the Round Robin algorithm at speed $\delta$. This way, we get $O(\delta^{-1})$-competitive algorithm in the worst case, because the schedule produced by Round Robin is 2-competitive with respect to the optimal schedule processed at speed $\delta$. Dinitz et al. [2022] used the same approach to robustify their algorithm, incurring the factor

---

[5]Formally, the loss of a permutation $\pi$ is the expected value of the following random variable: sample a pair of jobs uniformly at random; if the ordering of the jobs in $\pi$ places the longer job before the shorter one, output the difference between their respective lengths. Conversely, if the ordering in $\pi$ does not violate the length ordering, output zero. Notice that the optimal permutation has 0 loss and moreover expected loss of any permutation $\pi$ is proportional to the regret; that is, to the difference between the objective achieved by $\pi$ and the one achieved by the optimal permutation, as shown by Lindermayr and Megow [2022].

[6]If a solution for instance $I$ has total completion time objective $\mathrm{OPT}(I) + R$, then the same solution on a scaled instance $I'$ obtained from $I$ by multiplying all job lengths by $\alpha$ has objective $\alpha(\mathrm{OPT}(I) + R) = \mathrm{OPT}(I') + \alpha R$.

$\frac{1}{1-\delta}$ on top of their bound quoted above. This procedure unfortunately worsens the performance of the original algorithm by a constant factor, i.e., such robustification of our algorithm achieves additive regret only with respect to $\frac{1}{1-\delta}$ OPT$(I)$.

## 3 Related Work

The closest works to ours are by Dinitz et al. [2022] and Emek et al. [2021]. Dinitz et al. [2022] design algorithms with access to multiple predictors. They study (offline) min-cost bipartite matching, non-clairvoyant scheduling, and online load balancing on unrelated machines.[7] The main difference from our approach is conceptual: while we treat the task of identifying the best prediction as an independent modular learning problem, they treat it as an integral part of their algorithms. In the case of load balancing, they propose an $O(\log \ell)$-competitive algorithm which combines solutions of $\ell$ prediction-based algorithms into a fractional solution. A fractional solution can be rounded online, incurring an additional multiplicative factor of $\Theta(\frac{\log \log m}{\log \log \log m})$ where $m$ is the number of machines, see [Lattanzi et al., 2020, Li and Xian, 2021]. For non-clairvoyant scheduling for minimizing total completion time, they propose an algorithm which works under the assumption that no set of at most $\log \log n$ jobs has a large contribution to OPT. Their algorithm achieves a total completion time of $(1 + \epsilon)$ OPT $+O(\epsilon^{-5})\mu^*$ for any $\epsilon > 0$, where $\mu^*$ denotes the difference between the cost of the best available prediction and the cost of the offline optimum.

Emek et al. [2021] study caching with $\ell$ predictors which predict either the whole instance, or the next arrival time of the currently requested page. Based on each prediction, they build an algorithm with performance depending on the number of mistakes in this prediction. Then, they combine the resulting $\ell$ algorithms using the technique of Blum and Burch [2000] to build a single algorithm with a performance comparable to the best of them. Note that our approach is in a sense opposite to theirs: we use online learning techniques to build a single predictor comparable to the best of the given $\ell$ predictions and then we use this predictor in a simple algorithm. Their algorithm has a regret bound of $O(\mu^\star + k + \sqrt{Tk \log \ell})$, where $T$ is the length of the sequence, $k$ is the size of the cache, and $\mu^*$ is either the hamming distance of the actual input sequence from the closest predicted sequence or the number of mispredicted next arrival times in the output of the best predictor. This bound is larger than ours unless $T$ is very small, e.g., $o(k \log \ell)$.

There are numerous works on data-driven algorithm design, see the survey [Balcan, 2021]. They consider (potentially infinite) hypothesis classes containing parametrizations of various algorithms and utilize learning techniques to identify the best hypothesis given its performance on past data. The main difference from our work is that the hypothesis is chosen during the analysis of past data and before receiving the current input instance. In our case, learning happens as we receive larger and larger parts of the current input instance.

There are papers that consider our problems in a setting with a single black-box predictor which corresponds to our agnostic setting with a hypothesis class of size 1. For caching, these are [Lykouris and Vassilvitskii, 2021, Rohatgi, 2020, Wei, 2020, Antoniadis et al., 2023]. For online load balancing on unrelated machines and its special case restricted assignment, there are works on algorithms using predicted weights [Lattanzi et al., 2020, Li and Xian, 2021]. The papers [Purohit et al., 2018, Wei and Zhang, 2020, Im et al., 2021, Lindermayr and Megow, 2022] address the problem of non-clairvoyant scheduling.

Other related papers are by Bhaskara et al. [2020] who studied online linear optimization with several predictions, and [Anand et al., 2022, Antoniadis et al., 2023] who designed algorithms competitive with respect to a dynamic combination of several predictors for online covering and the MTS problem, respectively. There are also works on selecting the single best prediction for a series of input instances online [Khodak et al., 2022] and offline [Balcan et al., 2021]. The main difference from our work is that they learn the prediction before solving the input instance while we learn the prediction adaptively as we gain more information about the input instance.

---

[7]They state their result for a special case called restricted assignment, because no ML-augmented algorithms for unrelated machines were known at that time. However, they mention in the paper that their approach works also for unrelated machines.

Other relevant works are on various problems in online learning which consider switching costs [Cesa-Bianchi et al., 2013, Altschuler and Talwar, 2018] and on online smoothed optimization [Goel et al., 2019, Zhang et al., 2021, Chen et al., 2018].

Since the seminal papers of Lykouris and Vassilvitskii [2021] and Kraska et al. [2018], many works on ML-augmented algorithms appeared. There are by now so many of these works that is not possible to survey all of them here. Instead, we refer to the survey of Mitzenmacher and Vassilvitskii [2020] and to the website maintained by Lindermayr and Megow [2023].

Caching in offline setting was studied by Belady [1966]. Sleator and Tarjan [1985], laying the foundations of online algorithms and competitive analysis, showed that the best competitive ratio achievable by a deterministic caching online algorithm is $k$. Fiat et al. [1991] proved that the competitive ratio of randomized caching algorithms is $\Theta(\log k)$. Non-clairvoyant scheduling with the total completion time objective was studied by Motwani et al. [1993] who showed that Round-Robin algorithm is 2-competitive and that this is the best possible competitive ratio. Azar et al. [1992] proposed an $O(\log m)$-competitive algorithm for online load balancing on unrelated machines and showed that this is the best possible. In the offline setting, Lenstra et al. [1990] proposed a 2-approximation algorithm and this remains the best known algorithm. There was a recent progress on special cases [Svensson, 2012, Jansen and Rohwedder, 2017].

## 4    Warm-up: Caching in the Realizable Setting

In this section, we describe the simplest use case of our approach and that is caching in the realizable setting. In caching, we have a universe of pages $U$, a cache of size $k$ and its initial content $x_0 \in \binom{U}{k}$. As it is usual, we assume that $U$ contains $k$ "blank" pages $b_1, \ldots, b_k$ which are never requested and $x_0 = \{b_1, \ldots, b_k\}$, i.e., we start with an empty cache. We receive a sequence of requests $r_1, \ldots, r_T \in U \setminus \{b_1, \ldots, b_k\}$ online. At each time step $t$, we need to ensure that $r_t$ is present in the cache, i.e., our cache $x_t \in \binom{U}{k}$ contains $r_t$. If $r_t \notin x_{t-1}$ we say that there is a *page fault* and we choose $x_t \in \binom{U}{k}$ such that $r_t \in x_t$. This choice needs to be made without the knowledge of the future requests.

We measure the cost of a solution to a caching instance by counting the number of page loads (or, equivalently, page evictions) performed when transitioning from $x_{t-1}$ to $x_t$ at each time $t = 1, \ldots, T$. Denoting $d(x_{t-1}, x_t) = |x_t \setminus x_{t-1}|$, the total cost of the solution $x = x_1, \ldots, x_T$ is

$$\mathrm{cost}(x) = \sum_{t=1}^{T} d(x_{t-1}, x_t).$$

**Offline algorithm FitF.**    An intuitive offline optimal algorithm FitF was proposed by Belady [1966]: if there is a page fault at time $t$, it evicts a page from $x_{t-1}$ which is requested furthest in the future (FitF). In case there are pages which will never be requested again, it breaks the ties arbitrarily. The following monotonicity property will be useful in our analysis.

**Observation 4.** *Consider a request sequence $r_1, \ldots, r_T$. For any $t \leq T$, the cost incurred until time $t$ by FitF algorithm for sequence $r_1, \ldots, r_T$ is the same as the cost incurred by FitF algorithm for sequence $r_1, \ldots, r_t$.*

To see why this observation holds, it is enough to notice that the solution produced by FitF on $r_1, \ldots, r_T$ until time $t$ is the same as the solution of FitF on $r_1, \ldots, r_t$ which breaks ties according to the arrival times in $r_{t+1}, \ldots, r_T$.

**Learning task.**    In the realizable setting, we are given a class $\mathcal{H}$ of $\ell$ hypotheses $r^1, \ldots, r^\ell \in U^T$ such that the actual input sequence of requests $r$ is one of them (but we do not know which one). We split the task of designing an algorithm for this setting into two parts. First, we design an (improper) predictor that maintains a predicted sequence $\pi = \pi_1, \ldots, \pi_T$. This predictor makes a small number of switches (changes to $\pi$) until it determines the correct hypothesis. Second, we design an algorithm which uses an access to such predictor and its performance depends on the number of switches made by the predictor.

**Predictor.**    Our Predictor 1 below is based on a majority rule. It maintains a set $A$ of all hypotheses (sequences) in the class $\mathcal{H}$ which are consistent with the past requests. In time $t = 1$ the set $A$ is

initialized to be the entire class, i.e. $A = \mathcal{H}$, and it is updated whenever the current request $r_t$ differs from the predicted request $\pi_t$ (i.e., when there is a prediction error). The prediction $\pi$ used by our predictor is defined based on the set $A$ as follows: We set $\pi_t := r_t$ and, for $\tau = t+1, \ldots, T$, we choose $\pi_\tau$ to be the request agreeing with the largest number of hypotheses in $A$. This way, the predicted sequence $\pi$ is modified exactly after time-steps $t$ when $\pi_t \neq r_t$, and whenever this happens at least half of the hypotheses in $A$ are removed. Observe that we assume the realizable setting and hence at all times $A$ contains the hypothesis which is consistent with the input sequence. In particular $A$ is never empty. This implies the following lemma:

**Lemma 5.** *In realizable setting, Predictor 1 with a class $\mathcal{H}$ of $\ell$ hypotheses makes $\sigma \leq \log \ell$ switches and the final prediction is consistent with the whole input.*

---

**Predictor 1:** Majority predictor for caching in realizable setting

---

1 **for** $t = 1, \ldots, T$ **do**
2    **if** $t = 1$ *or prediction $\pi_t$ differs from the real request $r_t$* **then**      // make a switch
3      $A := \{ i \in \{1, \ldots, \ell\} \mid r_\tau^i = r_\tau \ \forall \tau = 1, \ldots t \}$ ;      // consistent hypotheses
4      update $\pi_t = r_t$ and $\pi_\tau = \arg\max_{p \in U} |\{i \in A \mid r_\tau^i = p\}|$ for each $\tau = t+1, \ldots, T$;

---

**Algorithm.** Our overall algorithm (See Algorithm 2) uses Predictor 1 and maintains the FitF solution $x_1, \ldots, x_T \in \binom{U}{k}$ for the current prediction $\pi$ at time $t$. Then it changes the cache to $x_t$. This solution needs to be recomputed whenever $\pi$ is modified.

---

**Algorithm 2:** caching, realizable setting

---

1 **for** $t = 1, \ldots, T$ **do**
2    **if** *there is a switch* **then**
3      receive $\pi$ from the predictor;
4      compute FitF solution $x_1, \ldots, x_T$ for $\pi$;
5    move to $x_t$.

---

**Lemma 6.** *Consider an input sequence $r$ and let $\mathrm{OPT}(r)$ denote the cost of the optimal offline solution for this sequence. Algorithm 2 with a predictor which makes $\sigma$ switches and its final prediction is consistent with $r$ incurs cost at most*

$$\mathrm{OPT}(r) + k\sigma.$$

Proof of Lemma 6 can be found in Appendix A. Combining lemmas 5 and 6, we get an algorithm for caching in a realizable setting with the following guarantee.

**Theorem 7.** *There is an algorithm for caching in realizable setting which, given a class $\mathcal{H}$ of $\ell$ hypotheses, achieves regret at most $k \log \ell$.*

## 5 Conclusions

We introduce a new approach to modeling algorithms with predictions. Unlike the traditional black-box access to a predictor, we extend the algorithmic problem by studying the accompanying learning problem. This allows the algorithm designer to improve the algorithm by:

1. Learning while processing the input,
2. Classifying the input instance to select the most suitable strategy before incurring high costs,
3. Accelerating the classification of the input by taking actions that may not be directly aligned with any strategy suggested by past data.

To achieve this, we split the computational problem into a learning component and an algorithmic component, addressing each separately. By applying our algorithms to existing settings with prediction portfolios [Dinitz et al., 2022, Emek et al., 2021], we demonstrate that our approach often results in simpler algorithms with improved performance compared to previous methods.

## Acknowledgements

Haim Kaplan is supported by Israel Science Foundation (ISF) grants 1595/19 and 1156/23, and by the Blavatnik Research Foundation.

Yishay Mansour has received funding from the European Research Council (ERC) under the European Union's Horizon 2020 research and innovation program (grant agreement No. 882396), by the Israel Science Foundation, the Yandex Initiative for Machine Learning at Tel Aviv University and a grant from the Tel Aviv University Center for AI and Data Science (TAD).

Shay Moran is a Robert J. Shillman Fellow; he acknowledges support by ISF grant 1225/20, by BSF grant 2018385, by an Azrieli Faculty Fellowship, by Israel PBC-VATAT, by the Technion Center for Machine Learning and Intelligent Systems (MLIS), and by the the European Union (ERC, GENERALIZATION, 101039692). Views and opinions expressed are however those of the author(s) only and do not necessarily reflect those of the European Union or the European Research Council Executive Agency. Neither the European Union nor the granting authority can be held responsible for them.

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

# Appendix

## A    Caching in realizable setting: omitted proof

*Proof of Lemma 6.* At every switch, we pay at most $k$ for switching the cache from the FitF solution of the previous predicted sequence to the cache of the newly computed one. Thus it suffices to show that in between these switches, our algorithm has the same number of page faults as OPT.

Denote $t_1, \ldots, t_\sigma$ the times when switches happen; for convenience, we also define $t_0 = 1$, $t_{\sigma+1} = T + 1$. We denote $\pi^0, \ldots, \pi^\sigma$ the predictions such that $\pi^{i-1}$ was predicted before the $i$th switch and $\pi^i$ after. Let $\kappa_i^j$ be the cost of the FitF solution for $\pi^j$ paid during time steps $t_i, \ldots, t_{i+1} - 1$, for $i = 0, \ldots, \sigma$. In this notation, the cost of OPT is $\sum_{i=0}^{\sigma} \kappa_i^\sigma$, since $\pi^\sigma = r$ (recall that $r$ is the input sequence), while, excluding the switching cost considered above, the cost of the algorithm is $\sum_{i=0}^{\sigma} \kappa_i^i$, because it pays $\kappa_i^i$ during time steps $t_i, \ldots, t_{i+1} - 1$ when following a FitF solution for $\pi^i$. We use induction on $i$ to show that $\kappa_i^j = \kappa_i^\sigma$ for each $j = i, \ldots, \sigma$. This implies $\sum_{i=0}^{\sigma} \kappa_i^\sigma = \sum_{i=0}^{\sigma} \kappa_i^i$. This essentially follows from Obsevation 4 and the fact that all sequences $\pi^j$, $j = i, \ldots, \sigma$ agree on the prefix up to time $t_{i+1} - 1$. The formal details follow.

In the base case with $i = 0$, Observation 4 implies that FitF solutions for each $\pi^i$ incur the same cost during time steps $1, \ldots, t_1 - 1$ and we have $\kappa_0^1 = \cdots = \kappa_0^\sigma$. For $i > 0$, the cost of the FitF solution for each $\pi^j$ with $j \geq i$ during time steps $1, \ldots, t_{i+1} - 1$ is $\sum_{m=0}^{i} \kappa_m^j = \sum_{m=0}^{i-1} \kappa_m^\sigma + \kappa_i^j$ by induction hypothesis. Using Observation 4, $\sum_{m=0}^{i-1} \kappa_m^\sigma + \kappa_i^i = \cdots = \sum_{m=0}^{i-1} \kappa_m^\sigma + \kappa_i^\sigma$ and therefore $\kappa_i^j = \kappa_i^\sigma$ for each $j = i, \ldots, \sigma$.

In total, the algorithm incurs cost

$$\mathrm{ALG}(r) \leq \sum_{i=0}^{\sigma} \kappa_i^i + \sigma k = \sum_{i=0}^{\sigma} \kappa_i^\sigma + \sigma k = \mathrm{OPT}(r) + \sigma k. \qquad \square$$

## B    Online Load Balancing on Unrelated Machines

We are given a set $M$ of $m$ machines and a sequence of jobs arriving online. At each time step, we receive some job $j$ described by a vector $p_j$, where $p_j(i)$ is its processing time on machine $i \in M$. We say that the job $j$ has *type* $p_j$. We have to assign job $j$ to one of the machines immediately without knowledge of the jobs which are yet to arrive. Our objective is to build a schedule of minimum *makespan*, i.e., denoting $J_i$ the set of jobs assigned to a machine $i$, we minimize the maximum load $\sum_{j \in J_i} p_j(i)$ over all $i \in M$. The best competitive ratio achievable in online setting is $\Theta(\log m)$ [5]. In offline setting, there is a 2-approximation algorithm.

**Proposition 8** (Lenstra et al. [26]). *There is an offline 2-approximation algorithm for load balancing on unrelated machines.*

Here we use the *competitive ratio* to evaluate the performance of our algorithms. We say that a (randomized) algorithm ALG achieves competitive ratio $r$ (or that ALG is $r$-competitive), if $\mathbb{E}[\mathrm{cost}(\mathrm{ALG}(I))] \leq r \cdot \mathrm{cost}(\mathrm{OPT}(I)) + \alpha$ holds for every instance $I$, where $\mathrm{OPT}(I)$ denotes the offline optimal solution for instance $I$ and $\alpha$ is a constant independent of $I$.

**Learning task.**    We are given a class $\mathcal{H}$ of $\ell$ hypotheses $H_1, \ldots, H_\ell$, each specifying the frequency $f_p \geq 0$ for every job type such that $\sum_p f_p = 1$. For simplicity, we assume that there is a constant $\delta > 0$ such that all frequencies in all hypotheses are integer multiples of $\delta$. For a hypothesis $H$ with frequencies $f_p$ and an integer $h \geq 1$, we define a scaling $H(h)$ as an instance containing $hf_p/\delta$ jobs of type $p$, for each $p$. This way, any type $p$ with $f_p \neq 0$ is represented by at least a single job in $H(1)$. We denote $c_0$ the largest makespan of $H(1)$ over all hypotheses $H \in \mathcal{H}$.

We say that an instance $I'$ is *consistent* with the input so far, if the following holds for every job type $p$: the number of jobs of type $p$ in instance $I'$ is greater or equal to the number of jobs of type $p$ which already arrived in the input sequence.

We note that the value $c$ of the makespan of the real instance can be guessed (up to a factor of two) by doubling while loosing only a constant factor in the competitive ratio. Using doubling and a

2-approximation offline algorithm (Proposition 8), we can also find a scaling $I_i$ of each hypothesis $H_i$ such that the makespan of $I_i$ is between $c$ and $4c$. Therefore, we start by assuming that we have the value of $c$ and the scaled instances $I_1, \ldots, I_\ell$ in advance, and postpone the discussion of finding them to Section B.3. We begin with the realizable setting, where the task of the predictor is either to identify an instance $I_i$ consistent with the whole input or return ERR which signals an incorrect value of $c$. Agnostic case is considered in Section B.4 and robustification of our algorithms in Section B.5. Section B.7 contains lower bounds.

## B.1 Predictors

We propose a deterministic and a randomized predictors for the realizable setting. Each of these predictors receives $c$ – the guessed value of the makespan of the real instance – and $\ell$ problem instances $I_1, \ldots, I_\ell$ whose makespan is between $c$ and $4c$ created by scaling the hypotheses $H_1, \ldots, H_\ell$. For each $p$ and $i$, we denote $n_p^i$ the number of jobs of type $p$ in instance $I_i$. Both predictors switch to a new prediction whenever they discover that, for some $p$, the number of jobs of type $p$ which arrived so far is already larger than their predicted number. At each switch, they update a set $A \subseteq \{1, \ldots, \ell\}$ of instances which are consistent with the input up to the current moment, i.e., the number of jobs of type $p$ which appeared so far is at most $n_p^i$ for all $p$ and $i \in A$.

There is a simple proper predictor which makes $\ell$ switches. This predictor starts by predicting according to an arbitrary instance. Whenever the current instance stops being consistent with the jobs arrived so far, it removes this instance from $A$, and switches to an arbitrary instance still in $A$. In what follows, we provide a more sophisticated predictors with lower number of switches.

**Deterministic predictor.** Our deterministic predictor is improper. At each switch, it predicts a "median" instance $\tilde{I}$ created from the instances in $A$ as follows. For each type $p$, $\tilde{I}$ contains $\tilde{n}_p$ jobs of type $p$, where $\tilde{n}_p$ is a median of $n_p^i$ over all $i \in A$. This way, whenever the number of jobs of type $p$ exceeds $\tilde{n}_p$, at least half of the instances are removed from $A$. Once $A$ is empty, the algorithm returns ERR. In the realizable setting, this happens when the guess of $c$ is not correct.

---

**Predictor 3:** load balancing on unrelated machines, deterministic

---

1   $A := \{1, \ldots, \ell\}$
2   choose $\tilde{n}_p$ as a median of $\{n_p^i \mid i = 1, \ldots, \ell\}$ for each job type $p$   // switch to initial $\tilde{I}$
3   **for** *time step when, for some $p$ the $(\tilde{n}_p + 1)$st job of type $p$ arrives* **do**
4      $A :=$ set of instances consistent with the input so far
5      **if** $A = \emptyset$ **then** return ERR
6      **for** *each job type $p$* **do**
7         choose $\tilde{n}_p$ as a median of $\{n_p^i \mid i \in A\}$.              // switch to a new $\tilde{I}$

---

**Lemma 9.** *Predictor 3 maintains a prediction which is always consistent with the jobs arrived so far. Given $\ell$ instances of makespan between $c$ and $4c$, it makes $\sigma \leq \log \ell$ switches before it either identifies an instance $I_{i^\star}$ consistent with the whole input, or returns ERR if there is no such instance. The makespan of each prediction is at most $4c \log \tau$, where $\tau$ is the number of distinct job types present in the instances $I_1, \ldots, I_\ell$.*

*Proof.* First, note that whenever a number of jobs of some type exceeds the prediction, Predictor 3 switches to a new prediction consistent with this number.

Consider a switch when the number of jobs of type $p$ exceeds $\tilde{n}_p$. Since $\tilde{n}_p$ was chosen as a median of $n_p^i$ over $i \in A$, the size of $A$ decreases by factor of at least 2 by this switch. Therefore, we have at most $\log \ell$ switches.

Now, we bound the makespan of $\tilde{I}$ by $4c \log \tau$, where $\tilde{I}$ is the prediction constructed from the current set of instances $A$. We do this by constructing a schedule for $\tilde{I}$. We say that an instance $I_i$ covers type $p$, if $n_p^i \geq \tilde{n}_p$. By the construction of $\tilde{I}$, we have that for every $p$, at least half of the instances in $A$ cover $p$. This implies that we can find an instance $I_i$ covering half of the job types. Indeed, consider a matrix $M$ whose columns correspond to instances $i$ in $A$ and its rows to $\tau' \leq \tau$ different job types

$p$ present in instances of $A$. We define $M_{p,i}$ as 1 if $I_i$ covers $p$ and 0 otherwise. Since every row has at least $|A|/2$ ones, there are at least $\tau'|A|/2$ ones in $M$ so there must be a column $i$ of $M$ containing $\tau'/2$ ones.

So, we pick $I_i$ and add all its jobs to the schedule, using a schedule of $I_i$ of makespan at most $4c$. Then we remove $i$ from $A$, and remove all job types covered by $I_i$ from $\tilde{I}$. $|A|$ decreases by 1, $\tau'$ decreases by factor of 2, and we still have that every remaining job type is covered by at least $|A|/2$ instances: This follows since for any type $p$ not covered by $I_i$, the number of instances covering it remains the same while $|A|$ decreases by 1. Therefore, after iterating this process at most $\log \tau$ times we cover all job types using makespan at most $4c \log \tau$. □

**Randomized predictor.** Our randomized predictor is proper. At each switch, it predicts an instance $I_i$, where $i$ is chosen from $A$ uniformly at random. We show that it satisfies the same bound on the number of switches as Predictor 3, but the makespan of its predictions is much smaller.

---

**Predictor 4:** load balancing on unrelated machines, randomized

1   $A := \{1, \dots, \ell\}$ choose $i \in A$ uniformly at random
2   set $\tilde{n}_p = n_p^i$ for each job type $p$              `// switch to initial` $\tilde{I}$
3   **for** *time step when, for some $p$ the $(\tilde{n}_p + 1)$st job of type $p$ arrives* **do**
4      $A :=$ set of instances consistent with the input so far
5      **if** $A = \emptyset$ **then** return ERR
6      choose $i \in A$ uniformly at random
7      set $\tilde{n}_p = n_p^i$ for each job type $p$         `// switch to a new` $\tilde{I}$

---

**Lemma 10.** *Predictor 4 maintains a prediction which is always consistent with the jobs arrived so far. Given $\ell$ instances of makespan between $c$ and $4c$, it makes $\sigma \leq \log \ell$ switches in expectation before it either identifies an instance $I_{i^\star}$ consistent with the whole input, or returns ERR if there is no such instance. Each prediction has makespan at most $4c$.*

*Proof.* First, note that whenever a number of jobs of some type exceeds the prediction, Predictor 4 switches to a new prediction consistent with this number.

Now, we bound the expected number of switches done by Predictor 4 in the style of the airplane seat problem. For every $i = 1, \dots, \ell$, define $t_i$ as the time step when an arriving job of type $p$ exceeds $n_p^i$, or $\infty$ if no such time step exists. Note that for every instance $i$ which is inconsistent with the input, $t_i$ is finite. The instances are eliminated in the order of increasing $t_i$. Since we choose $i$ uniformly at random over the remaining instances, we have $t_j \leq t_i$ for at least half of the inconsistent instances (in expectation). The formal proof then follows from the traditional analysis of the lost boarding pass problem, see e.g. [20]. □

## B.2   Algorithm

Our algorithm uses a predictor whose predictions are always consistent with the jobs arrived so far. At each switch, it computes a 2-approximation of the optimal solution for the predicted instance using the algorithm from Proposition 8 and schedules jobs based on this solution until the next switch.

---

**Algorithm 5:** load balancing on unrelated machines

1   **for** *each incoming job $j$* **do**
2      **if** *this is the first job or there was a switch* **then**
3          get a new prediction $\tilde{I}$ from the predictor or return ERR
4          compute $Lenstra(\tilde{I})$
5      assign $j$ to a machine based on $Lenstra(\tilde{I})$

---

**Lemma 11.** *Given a predictor which makes $\sigma$ switches and produces predictions that are always consistent with the jobs arrived so far and have makespan at most $\kappa$, Algorithm 5 uses makespan at*

*most $2\kappa\sigma$ and either schedules all jobs in the input sequence or reports ERR if none of the instances is consistent with the input sequence.*

*Proof.* At each switch, Algorithm 5 starts building a new schedule for the predicted instance with makespan at most $2\kappa$. Therefore the total makespan is at most $2\kappa\sigma$. $\square$

### B.3 Guessing the Optimal Makespan and Scaling

First, we discuss how to find a scaling of a hypothesis $H$ that has the required makespan. Let $c$ be our estimate of OPT such that $c \leq \text{OPT} \leq 2c$. We start with $h = 1$ and keep doubling $h$ until $Lenstra(H(h))$ becomes at least $2c$. (and at most $4c$). Since Lenstra is a 2-approximation, we know that when $Lenstra(H(h)) \geq 2c$ then $\text{OPT}(H(h)) \geq c$. Since this is the smallest $h$ for which $Lenstra(H(h)) \geq 2c$ we know that $\text{OPT}(H(h/2)) \leq 2c$ so $\text{OPT}(H(h)) \leq 4c$.

Algorithm 6 is our scheduling algorithm. We start with the initial guess of $c_0$ for the optimal solution, where $c_0$ is an upper bound on the makespan of $H(1)$ for each $H \in \mathcal{H}$. At each iteration we double our guess $c$. We scale the hypotheses to build instances with makespan between $c$ and $4c$. We run Algorithm 5 with these instances until it returns error. We keep iterating until the whole input is processed.

---

**Algorithm 6:** guessing the optimal makespan by doubling

---
1 **for** $c = c_0, \ldots, 2^1 c_0, 2^2 c_0, 2^3 c_0, \ldots$ **do**
2     **for** $i = 1, \ldots, \ell$ **do**
3        find scaling $h_c^i$ such that $I_i := H_i(h_c^i)$ has makespan between $c$ and $4c$
4     Run Algorithm 5 with $I_1, \ldots, I_\ell$
5     **if** *not ERR* **then** finish: all the jobs are scheduled

---

**Lemma 12.** *If Algorithm 6 uses makespan at most $\gamma c$ in an iteration with guess $c$, then it uses makespan at most $O(\gamma)c^\star$, where $c^\star$ denotes the value of $c$ in the last iteration.*

*Proof.* The total makespan used by the algorithm is at most

$$\sum_{i=0}^{\log(c^\star/c_0)} 2^i c_0 \gamma \leq 2c^\star \gamma. \qquad \square$$

**Lemma 13.** *Let $c^\star$ be the value of $c$ in the last iteration of Algorithm 6 in the realizable setting. Then the makespan of the offline optimal solution is at least $c^\star/2$.*

*Proof.* Let $H_{i^\star}$ be the correct hypothesis describing the input instance $I$. We know that $H_{i^\star}(h_{c^\star/2}^{i^\star}) \subsetneq I$, otherwise Algorithm 6 would terminate in the previous iteration. We have

$$c^\star/2 \leq \text{OPT}(H_{i^\star}(h_{c^\star/2}^{i^\star})) \leq \text{OPT}(I),$$

implying $\text{OPT}(I) \geq c^\star/2$. The first inequality is by the choice of $h_{c^\star/2}^{i^\star}$ and the last one since $H_{i^\star}(h_{c^\star/2}^{i^\star}) \subsetneq I$. $\square$

**Theorem 14.** *There are algorithms for the realizable setting with deterministic and randomized predictors which, given a hypothesis class $\mathcal{H}$ of size $\ell$, achieve competitive ratio $O(\log \ell \log \tau)$ and $O(\log \ell)$ respectively, where $\tau$ is the total number of different job types in $\mathcal{H}$.*

*Proof.* Combining Lemmas 12 and 13, the makespan achieved by Algorithm 6 is at most $O(\gamma \text{ OPT})$. By Lemmas 9, 10, and 11, we have $\gamma = \log \ell \log \tau$ in case of the deterministic Predictor 3 and $\gamma = \log \ell$ in case of the randomized Predictor 4. $\square$

## B.4 Agnostic Setting

The algorithm above works also in the agnostic setting. Let $f_p$ and $f_p^\star$ be the frequency of job type $p$ according to a hypothesis $H \in \mathcal{H}$ and its true frequency, respectively. If there is a job type $p$ such that $f_p \neq 0$ and $f_p^\star = 0$, we define $\alpha_H := n + 1$, where $n$ denotes the number of jobs in the input instance. Otherwise, we define $\alpha_H := \max\{f_p/f_p^\star \mid f_p^\star \neq 0\}$. Similarly, if there is a job type $p$ such that $f_p = 0$ and $f_p^\star \neq 0$, we define $\beta_H := n + 1$. Otherwise, $\beta_H := \max\{f_p^\star/f_p \mid f_p \neq 0\}$. Note that both $\alpha_H$ and $\beta_H$ are at most $n + 1$: the smallest $f_p^\star > 0$ is at least $1/n$ for the job types represented by a single job and the same holds for the scaling $H(1)$ of each hypothesis $H \in \mathcal{H}$. Let $H \in \mathcal{H}$ be a hypothesis achieving the smallest product $\alpha_H \beta_H$. We call a pair $(\alpha, \beta)$, where $\alpha = \alpha_H$ and $\beta = \beta_H$ the error of hypothesis class $\mathcal{H}$.

We prove the following variant of Lemma 13 for agnostic setting, in the case with $\alpha, \beta \leq n$.

**Lemma 15.** *Let $c^\star$ be the value of $c$ in the last iteration of Algorithm 6 in the agnostic setting, given a hypothesis class $\mathcal{H}$ with error $\alpha, \beta \leq n$. Then, the makespan of the offline optimal solution is at least $\frac{c^\star}{O(\alpha\beta)}$.*

*Proof.* The main idea here is that even if all hypothesis are incorrect, Algorithm 6 terminates once the input instance $I$ is subsumed by a large enough scaling of some hypothesis. Consider the correct hypothesis $H^\star$ for $I$ consisting of real frequencies $f_p^\star$ for each job type $p$ ($H^\star$ may not be in $\mathcal{H}$ in the agnostic setting) and the best hypothesis $H \in \mathcal{H}$ (such that $\alpha, \beta = \alpha_H, \beta_H$). We assume below that $\alpha$ and $\beta$ are integers, otherwise we round them up to the closest integers.

We have the following: $I = H^\star(h)$ for some integer scaling $h$. Since, $f_p^\star \leq \beta f_p$, we have $H^\star(h) \subseteq H(\beta h)$. Therefore, in the last iteration of Algorithm 6, we have $c^\star$ within a constant factor from the optimum makespan of $H(\beta h)$ Similarly, since $f_p \leq \alpha f_p^\star$, we have that $H(\beta h) \subseteq H^\star(\alpha\beta h)$ which implies that the optimum makespan of $H(\beta h)$ is at most $\alpha \, \mathrm{OPT}(H^\star(\beta h)) \leq \alpha\beta \, \mathrm{OPT}(I)$. Altogether, we have $c^\star \leq O(\alpha\beta \, \mathrm{OPT}(I))$. $\qquad\square$

**Theorem 16.** *There are algorithms for the agnostic setting with deterministic and randomized predictors which, given a hypothesis class $\mathcal{H}$ of size $\ell$ with error $(\alpha, \beta)$, achieve competitive ratio $O(\alpha\beta \log \ell \log \tau)$ and $O(\alpha\beta \log \ell)$, respectively, where $\tau$ is the total number of different job types in $\mathcal{H}$.*

*Proof.* Consider an algorithm which schedules all jobs of type $p$ such that $f_p = 0$ according to all hypotheses in $\mathcal{H}$ to the machine $\arg\min_{i \in [m]} p_i$, i.e., the machine which can process the job fastest. Let $J_0$ denote the set of such jobs. All other jobs in $I \setminus J_0$ are scheduled using Algorithm 6. The resulting makespan is at most

$$|J_0| \, \mathrm{OPT} + \gamma c^* \leq |J_0| \, \mathrm{OPT} + O(\alpha\beta\gamma) \, \mathrm{OPT}$$

where $c^*$ is the value of $c$ in the last iteration of Algorithm 6 processing jobs in $I \setminus J_0$. This is because, for every $j \in J_0$, we have $\mathrm{OPT} \geq \min_{i \in [m]} p_i$. The inequality follows from lemmas 15 and 12.

If $|J_0| > 0$, then at least one of $\alpha$ and $\beta$ is at least $n + 1$. Therefore, our makespan is always at most $\alpha\beta\gamma \, \mathrm{OPT}$. By lemmas 9, 10, and 11, we have $\gamma = \log \ell \log \tau$ in case of the deterministic Predictor 3 and $\gamma = \log \ell$ in case of the randomized Predictor 4. $\qquad\square$

## B.5 Achieving Robustness

With large $\alpha, \beta$, the competitive ratio in Theorem 16 might be worse than $O(\log m)$ which is achievable without predictions, i.e., Algorithm 6 is not robust. This can be fixed easily: once Algorithm 5 returns ERR at iteration $c$, we run a classical online algorithm by Azar et al. [5] as long as it uses makespan $\gamma c$. That is, we stop it as soon as its makespan go above $\gamma c$ (and we do not schedule the job that makes it go above $\gamma c$. This increases the makespan of the solution by factor at most 2 and ensures that $\mathrm{OPT} \geq \gamma c/O(\log m)$.

The following lemma holds both in realizable and in the agnostic setting.

**Lemma 17.** *The makespan of the solution produced by Algorithm 7 is at most a constant factor higher than of Algorithm 6. Moreover, its competitive ratio is always bounded by $O(\log m)$.*

**Algorithm 7:** robust variant of Algorithm 6

---

**1** **for** $c = c_0, \ldots, 2^1 c_0, 2^2 c_0, 2^3 c_0, \ldots$ **do**
**2**     **for** $i = 1, \ldots, \ell$ **do**
**3**        find scaling $h_c^i$ such that $I_i := H_i(h_c^i)$ has makespan between $c$ and $4c$
**4**     Run Algorithm 5 with $I_1, \ldots, I_\ell$
**5**     Run Online algorithm of Azar et al. [5] as long as it uses makespan of at most $\gamma c$
**6**     **if** *all jobs are scheduled* **then** finish

---

*Proof.* Algorithm 6 terminates once it finds $c^\star$ and $i$, such that the actual instance $I$ is a subset of $H_i(h_{c^\star}^i)$. With the same $c^\star$, Algorithm 7 terminates as well. While Algorithm 6 uses a makespan of at most $\gamma c^\star$ in each iteration, Algorithm 7 uses makespan of at most $2\gamma c^\star$ in each iteration.

Now we prove the $O(\log m)$ bound on the competitive ratio. Consider $I' \subseteq I$ the set of jobs assigned to machines by the online algorithm of Azar et al. [5] in the next to last iteration (line 5 of Algorithm 7). We have $\mathrm{OPT}(I) \geq \mathrm{OPT}(I') \geq \frac{\gamma c^\star/2}{O(\log m)}$ because the online algorithm is $O(\log m)$-competitive and it required makespan $\gamma c^\star/2$ in the second to last iteration. Since Algorithm 7 uses makespan at most $O(\gamma c^\star)$ by Lemma 12, the bound on its competitive ratio follows. $\square$

### B.6   A note on combining arbitrary integral algorithms

Dinitz et al. [15] considered a portfolio of $\ell$ algorithms for load balancing on unrelated machines and proposed a way to combine their outputs in a single fractional solution of cost at most $O(\log \ell)$-times higher than the cost of the best algorithm in the portfolio. Such solution can be rounded online only by loosing an additional factor of $\Theta(\log\log m / \log\log\log m)$.

Our approach described above can be used to produce directly an integral solution as far as all the algorithms in the portfolio are integral. The cost of this solution is at most $O(\log \ell)$-times higher than the cost of the best algorithm in the portfolio.

We guess the value $c$ of the makespan achieved by the best algorithm in the potfolio using the doubling trick, loosing a constant factor due to this guessing as in Section B.3. We create a randomized predictor similar to Predictor 4 as follows. Start with the set of active algorithms $A := \{1, \ldots, \ell\}$ and predict an algorithm chosen from $A$ uniformly at random. Once its makespan exceeds $c$, we update $A$ to include only those algorithms whose current makespan is at most $c$, choose one of them uniformly at random and iterate. We continue either until all jobs are scheduled or until $A$ is empty which signals an incorrect guess of $c$. At each time step, we schedule the current job based on a decision of the algorithm currently chosen by the predictor, paying at most $c$ while following a single algorithm. An argument as in the proof of Lemma 10 shows that our predictor switches $O(\log \ell)$ algorithms in expectation at each iteration.

### B.7   Lower Bound

Our lower bound holds for a special case of load balancing on unrelated machines called *restricted assignment*, where processing of each job $j$ is restricted to a subset $S_j$ of machines, i.e., its processing time is 1 on all machines belonging to $S_j$ and $+\infty$ otherwise. Our construction requires $m \geq \ell$ machines and is inspired by the construction of Azar et al. [5] ($\ell$ is number of hypothesis in $\mathcal{H}$ as before). Since we can ensure that all jobs have infinite processing time on machines $\ell + 1, \ldots, m$, we can assume that $\ell = m$ and that $\ell$ is a power of two.

We construct $\ell$ instances of restricted assignment on $\ell$ machines, each with makespan $c \in \mathbb{N}$. In instance $i \in \{1, \ldots, \ell\}$, there are $c\ell/2^j$ jobs restricted to machines whose index agrees with $i$ in the $j - 1$ most significant bits, for $j = 1, \ldots, \log \ell$. In particular, each instance starts with $c\ell$ jobs which can be processed on any machine. The jobs arrive in iterations from $j = 1$ to $\log \ell$ (from less restricted to more restricted). If numbers $i$ and $i'$ have a common prefix of length $j - 1$, then instances $i$ and $i'$ have the same jobs in the first $j$ iterations.

Optimal solution for instance $i$ can be described as follows: For each $j = 1, \ldots, \log \ell$, schedule all $c\ell/2^j$ jobs evenly on machines whose index agrees with $i$ up to bit $j - 1$ but disagrees with $i$ in bit $j$:

There are $\ell/2^{j-1} - \ell/2^j = \ell/2^j$ such machines. We leave all the machines which agree with $i$ in the first $j$ bits empty for the following iterations. Since, for each $j$, we schedule $c\ell/2^j$ jobs evenly on $\ell/2^j$ machines, their load is $c$.

**Theorem 18.** *There is no (randomized) algorithm which, with a hypothesis class of size $\ell \leq m$, that achieves competitive ratio $o(\log \ell)$.*

*Proof.* The adversary chooses the correct instance $i$ bit by bit, fixing the $j$th bit $i_j$ at the end of iteration $j$ depending on the behavior of the algorithm. Bit $i_j$ is chosen according to the following procedure: Given the knowledge of the distribution over algorithm's decisions, count the expected number of jobs from iterations $1, \ldots, j$ assigned to machines whose first $j$ bits are $i_1, \ldots, i_{j-1}, 0$. If this number is higher than the expected number of jobs assigned to machines with prefix $i_1, \ldots, i_{j-1}, 1$, then choose $i_j = 0$. Otherwise, choose $i_j = 1$.

For each $j = 1, \ldots, \log \ell$, we denote $M_j$ the set of machines with prefix $i_1, \ldots, i_j$, with $M_0 = \{1, \ldots, \ell\}$. We show by induction on $j$ that at least $\frac{1}{2} j c\ell/2^j$ jobs are assigned to the machines belonging to $M_j$ in expectation. This way, $M_{\log \ell}$ contains a single machine with expected load at least $\frac{1}{2} c \log \ell$.

The base case $j = 1$ of the induction holds: We assign $c\ell$ jobs to $\ell$ machines in $M_0$ and $i_1$ is chosen so that machines in $M_1$ get at least half of them, in expectation. For $j > 1$, the expected number of jobs from iterations up to $j - 1$ assigned to machines in $M_{j-1}$ is at least $\frac{1}{2}(j-1)c\ell/2^{j-1}$ by the induction hypothesis. There are $c\ell/2^j$ jobs restricted to machines in $M_{j-1}$ scheduled in iteration $j$. Therefore, the total expected number of jobs from iterations $1, \ldots, j$ assigned to machines in $M_{j-1}$ is at least

$$\frac{1}{2}(j-1)c\frac{\ell}{2^{j-1}} + c\frac{\ell}{2^j} = jc\frac{\ell}{2^j}.$$

Since $i_j$ is chosen such that the machines in $M_j$ are assigned at least half of the jobs assigned to machines in $M_{j-1}$ in expectation, the expected number of jobs assigned to $M_j$ is at least $\frac{c}{2} j\ell/2^j$.

While the makespan for the instance $i$ is $c$, the machine in $M_{\log \ell}$ has expected load at least $\frac{1}{2} c \log \ell$, showing that the competitive ratio of the algorithm is at least $\frac{1}{2} \log \ell$. □

## C   Non-clairvoyant Scheduling

We have a single machine and $n$ jobs available from time $0$ whose lengths $p_1, \ldots, p_n$ are unknown to the algorithm. We know the length $p_j$ of the job $j$ only once it is finished. If it is not yet finished and it was already processed for time $x_j$, we only know that $p_j \geq x_j$. Our objective is to minimize the sum of the completion times of the jobs. To avoid scaling issue in our regret bounds, we assume that the length of each job is at most $1$. Note that if a solution for instance $I$ has total completion time objective $\mathrm{OPT}(I) + R$, then the same solution on a scaled instance $I'$ obtained from $I$ by multiplying all job lengths by $\alpha$ has objective $\alpha(\mathrm{OPT}(I) + R) = \mathrm{OPT}(I') + \alpha R$. There is a 2-competitive Round Robin algorithm which runs all unfinished jobs simultaneously with the same rate [34]. Consider an algorithm which schedules the jobs in order $1, \ldots, n$, denoting $p_1, \ldots, p_n$ their lengths. Then, its total completion time objective can be expressed as

$$\sum_{j=1}^{n} \sum_{i=1}^{j} p_j = \sum_{j=1}^{n} p_j(n - j + 1).$$

This objective is minimal if $p_1 \leq \cdots \leq p_n$ which is the ordering chosen by the optimal algorithm Shortest Job First (SJF) for the clairvoyant setting where we know lengths of all the jobs in advance [34].

**Learning task.**   We are given a class $\mathcal{H}$ of $\ell$ hypotheses, each of them specifies length of all jobs, denoting $p_j^i$ the length of job $j$ according to the hypothesis $H_i$. A predictor uses $\mathcal{H}$ to produce prediction $\pi$, where $\pi_j$ is the predicted length of the job $j$. We call a predictor *monotone* if, at each time step, it maintains a prediction which is consistent with our knowledge about job lengths and $\pi_j \leq p_j$ holds for every job $j$ (i.e., it never overestimates a length of a job). We propose a monotone predictor only for the realizable setting. Non-clairvoyant scheduling in the agnostic setting is considered in Section C.2 with a different kind of hypotheses.

**Predictor.** We propose a monotone predictor which works as follows. At the beginning, we start with $A := \{1, \ldots, \ell\}$. At each time instant $t$, we remove from $A$ each hypothesis $i$ such that there is some job $j$ which was already processed for time $x_j > p_j^i$. Whenever $A$ changes, we *switch* to a new prediction by updating the predicted lengths of unfinished jobs as follows. For every unfinished job, we predict the smallest length specified by any instance in $A$.

---

**Predictor 8:** non-clairvoyant scheduling, realizable setting

---

1  **for** $t = 0$ *or whenever some hypothesis is removed from $A$* **do**
2     $U :=$ set of unfinished jobs
3     **for** $j \in U$ **do** $\pi_j := \min\{p_j^i \mid i \in A\}$   `// switch: update pred. unfinished jobs`

---

**Lemma 19.** *In the realizable setting, Predictor 8 is monotone and makes $\sigma \leq \ell$ switches.*

*Proof.* Switch happens whenever $x_j = \pi_j$ for some unfinished job $j$. In that case, the hypothesis predicting $\pi_j$ for job $j$ is removed from $A$; therefore there can be at most $\ell$ switches.

In the realizable setting, there is a hypothesis $i^\star$ which is correct and is never removed from $A$. Therefore, at each time instant, we have $\pi_j \leq \pi_j^{i^\star} = p_j$ for any job $j$.     □

**Algorithm.** At each time instant, our algorithm receives the newest prediction from the predictor and always processes the job whose current predicted length is the smallest. When a switch happens, it interrupts the processing of the current job, leaving it unfinished.

---

**Algorithm 9:** non-clairvoyant scheduling

---

1  **for** *each time instant $t$* **do**
2     $U :=$ set of unfinished jobs
3     get the newest prediction $\pi$ from the predictor
4     run job $j := \arg\min_{j \in U}\{\pi_j\}$

---

**Lemma 20.** *With a monotone predictor which makes $\sigma$ switches, Algorithm 9 produces a schedule with total completion time at most $\mathrm{OPT}(I) + \sigma\sqrt{2\,\mathrm{OPT}(I)}$ on an input instance $I$ with job lengths bounded by $1$ and offline optimal completion time of $\mathrm{OPT}(I)$.*

*Proof.* We relabel the jobs so that $p_1 \leq \cdots \leq p_n$. The optimal solution is to schedule them in this exact order, always running the shortest unfinished job, achieving total completion time

$$\mathrm{OPT}(I) = \sum_{i=1}^{n} p_i \cdot (n - i + 1).$$

At each switch of the predictor, our algorithm leaves the current job unfinished. On the other hand, whenever a job $j$ is completed, it must have been the shortest unfinished job, because it was the unfinished job with the shortest $\pi_j$ and we have $p_j \leq \pi_j \leq \pi_{j'} \leq p_{j'}$ for any unfinished job $j'$ by the monotony of the predictor. Therefore, the total completion time of the algorithm is

$$\mathrm{ALG}(I) \leq \sum_{i=1}^{n} (C_i + p_i) \cdot (n - i + 1),$$

where $C_i$ is the time between the completion of jobs $i - 1$ and $i$ spent processing jobs which were left unfinished due to a switch – we denote the set of these jobs by $Q_i$. Algorithm 9 processes a job $j$ only when $\pi_j \leq \pi_{j'}$ for all unfinished jobs $j'$. Therefore, each job $j \in Q_i$ can contribute to $C_i$ at most $\pi_j \leq \pi_i \leq p_i$, by the monotony of the predictor. Therefore we can bound the cost of the algorithm as follows:

$$\mathrm{ALG}(I) = \mathrm{OPT}(I) + \sum_{i=1}^{n} C_i \cdot (n - i + 1) \leq \sum_{i=1}^{n} \sum_{j \in Q_i} p_i (n - i + 1).$$

Note that $\sigma = \sum_{i=1}^{n} |Q_i|$. So, the sum in the right-hand side contains $\sigma$ summands and each of them can be bounded by $\sqrt{2\,\mathrm{OPT}(I)}$, since we have

$$\left(p_i(n-i+1)\right)^2 \leq 2p_i \frac{(n-i+1)^2}{2} \leq 2\sum_{k=i}^{n} p_i(n-k+1) \leq 2\sum_{k=i}^{n} p_k(n-k+1) \leq 2\,\mathrm{OPT}(I).$$

The first inequality follows since $p_i \leq 1$ and the third inequality since $p_i \leq p_k$ for each $k \geq i$. Therefore, we have

$$\mathrm{ALG}(I) - \mathrm{OPT}(I) \leq \sigma\sqrt{2\,\mathrm{OPT}(I)}. \qquad \square$$

Lemma 19 and Lemma 20 imply the following theorem.

**Theorem 21.** *Consider an instance $I$ with maximum job length 1 and let $OPT(I)$ be the offline optimal completion time of $I$. There is an algorithm for the realizable setting which, given a hypothesis class $\mathcal{H}$ of size $\ell$, achieves objective value at most $\mathrm{OPT}(I) + \ell\sqrt{2\,\mathrm{OPT}(I)}$.*

### C.1 Instances with Two Distinct Lengths

Consider the case in which the larger jobs have length 1 and the smaller ones have length $\lambda \in [0, 1)$. We propose the following predictor which makes only $\log \ell$ switches and constructs its prediction based on the majority rule.

---

**Predictor 10:** non-clairvoyant scheduling with two distinct lengths

---

1 **for** *time instant $t$* **do**
2    **if** $t = 0$ *or some prediction was shown to be wrong* **then**
3      $A :=$ set of instances consistent with the input so far
4      $U :=$ set of unfinished jobs
5      **for** $j \in U$ **do** $\pi_j := \arg\max_{x=1,\lambda} |\{p_j^i = x \mid i \in A\}|$    // majority prediction

---

By its definition, Predictor 10 makes a switch every time its prediction is shown to be incorrect. The following lemma bounds its total number of switches.

**Lemma 22.** *Predictor 10 makes makes at most $\log \ell$ switches in total.*

*Proof.* When a prediction $\pi_j$ is shown to be incorrect, the predictor makes a switch and the size of $A$ decreases by at least factor of 2, because the length of $j$ was predicted to be $\pi_j$ by at least half of the hypotheses in $A$. Therefore, there can be at most $\log \ell$ switches. $\qquad \square$

Our algorithm works as follows. If there is an unfinished job $j$ with $\pi_j = \lambda$, it runs it to completion. Otherwise, it chooses an unfinished job predicted to have length 1 uniformly at random and runs it to completion. The following lemma is useful for the analysis.

---

**Algorithm 11:** non-clairvoyant scheduling with two distinct lengths

---

1 **for** *time step $0$ and whenever some job is finished* **do**
2    update the prediction $\pi$
3    $U :=$ set of unfinished jobs
4    **if** *there is $j \in U$ s.t. $\pi_j = \lambda$* **then** run $j$ until it is completed
5    **else** choose $j$ from $U$ uniformly at random and run it to completion

---

**Lemma 23.** *Consider two schedules without preemption such that the second one differs from the first one by moving a single job of size 1 earlier, jumping over $d$ jobs. Then the total completion time of the second schedule is larger by at most $d - \sum_{i=1}^{d} p_i$ where $p_1, \ldots, p_d$ are the processing times of the jobs which were delayed. This difference is at most $(1 - \lambda)d$. If all these $d$ jobs have length $\lambda$, then the difference is exactly $(1 - \lambda)d$.*

*Proof.* The completion time of the job we shifted earlier get smaller by $\sum_{i=1}^{d} p_i$, and the completion time of the $d$ jobs which were delayed increases by at most 1. All other completion times do not change. $\qquad\square$

**Lemma 24.** *Algorithm 11 for instances with job lengths in $\{1, \lambda\}$ with a predictor which makes a switch whenever its prediction is shown to be incorrect and makes $\sigma$ switches in total produces a schedule with expected total completion time at most $\mathrm{OPT}(I) + \sigma(1 - \lambda)n$, where $\mathrm{OPT}(I)$ is the total completion time of the offline optimal solution.*

*Proof.* The offline optimum schedules all jobs of length $\lambda$ before the jobs of length 1. If we finish a job and there was no switch, the prediction of its length was correct. We write $\sigma = \sigma' + \sigma''$, where $\sigma'$ is the number of times we process a job with incorrect predicted length $\lambda$ (type-1 switch) and $\sigma''$ is the number of times we process a job with incorrect predicted length 1 (type-2 switch).

Every type-1 switch causes a job of length 1 to be scheduled before at most $n$ jobs of length $\lambda$. By Lemma 23, the schedule produced by the algorithm is more expensive than the solution where these $\sigma'$ jobs were executed last by at most $\sigma'(1 - \lambda)n$. It remains to analyze by how much this modified schedule, in which type-1 switches never happen, is more expensive than the optimal schedule.

We split the time horizon into intervals moments when a type-2 switch happens (recall that the switch happens right after we scheduled a short job with predicted length 1). There are $\sigma'' + 1$ intervals $i = 0, 1, \ldots, \sigma''$. We denote by $q_i$ the number of jobs of predicted length $\lambda$ scheduled first in the $i$th interval (including the first job causing the type-2 switch) and by $m_i$ the number of jobs of predicted length 1 scheduled thereafter in the $i$th interval. Let $n_i$ denote the total number of unfinished jobs when we finish scheduling the $q_i$ jobs of predicted length $\lambda$, and let $s_i$ be the number of unfinished jobs of length $\lambda$ at that time. We have $q_i = s_{i-1} - s_i$.

With this notation and using Lemma 23, the regret of the algorithm is

$$(1 - \lambda) \sum_{i=0}^{\sigma''-1} m_i s_i$$

This is because the optimal schedule processes jobs of length $\lambda$ first and our schedule can be constructed by moving (one by one) $m_i$ jobs of length 1 forward, leaving $s_i$ jobs of length $\lambda$ behind for each $i = 1, \ldots, \sigma''$.

Since we choose to process a random job of predicted length 1 we have $\mathbb{E}[m_i \mid n_i, s_i] = \frac{n_i+1}{s_i+1} \leq \frac{n}{s_i}$, as we are drawing from $n_i$ jobs without replacement until the first of $s_i$ jobs of size $\lambda$ is drawn. Therefore $\mathbb{E}[m_i \mid s_i] \leq n/s_i$ and $\mathbb{E}[m_i s_i] = \sum_{j=1}^{n} \mathbb{P}(s_i = j)\mathbb{E}[m_i s_i \mid s_i = j] \leq n$. So, the expected regret in case a type-1 switch never happens is

$$(1 - \lambda) \sum_{i=0}^{\sigma''-1} \mathbb{E}[m_i s_i] \leq (1 - \lambda)n\sigma''.$$

Therefore, the cost of the algorithm is $\mathrm{ALG} \leq \mathrm{OPT} + (1 - \lambda)n\sigma$. $\qquad\square$

Lemma 22 and Lemma 24 together imply the following theorem.

**Theorem 25.** *Consider an instance $I$ with jobs of length either 1 or $\lambda$ for some fixed $\lambda \in (0, 1)$. There is an algorithm which, given a hypothesis class $\mathcal{H}$ of size $\ell$, produces a schedule with expected total completion time at most $\mathrm{OPT}(I) + \sigma(1 - \lambda)n$ in the realizable setting (i.e., $I \in \mathcal{H}$), where $\mathrm{OPT}(I)$ is the total completion time of the offline optimal solution.*

### C.2 Agnostic Setting

We propose an algorithm for the agnostic setting with a different type of hypotheses, each specifying an optimal ordering of the jobs rather than their lengths. Given a class of such hypotheses $\mathcal{H}$, the predictor maintains an ordering $\pi$ and, at each switch, it can change the ordering of the unfinished jobs. Let $J = \{1, \ldots, n\}$ be the set of all jobs. We call a *mistake* every inversion in this ordering, i.e., two jobs $i, j \in J$ such that $p_i < p_j$ but $\pi(j) < \pi(i)$. For every pair of jobs $\{i, j\}$, we define

$$\mu(\pi, \{i, j\}) = \begin{cases} (p_i - p_j)^+ & \text{if } \pi(i) < \pi(j) \\ (p_j - p_i)^+ & \text{otherwise.} \end{cases}$$

If the order of $i$ $j$ in $\pi$ is incorrect, then $\mu(\pi, \{i,j\})$ is the weight of this mistake. Otherwise, it is equal to 0. For a set of pairs of jobs $P \subseteq \binom{J}{2}$, where $\binom{J}{2}$ denotes the set of all pairs of jobs, we denote $\mu(\pi, P) = \sum_{\{i,j\}\in P} \mu(\pi, \{i,j\})$, and $\mu(\pi) = \mu(\pi, \binom{J}{2})$ is the total weight of mistakes in $\pi$.

**Proposition 26** (Lindermayr and Megow [28])**.** *Let* OPT *denote the cost of the offline optimal solution and* $\mathrm{cost}(\pi)$ *the cost of the solution where the jobs are processed according to the ordering* $\pi$. *Then* $\mathrm{cost}(\pi) - \mathrm{OPT} = \mu(\pi)$.

**Predictor.** It has a parameter $m$. First, it samples $m$ pairs of jobs, $P = \{\{j_i, j_i'\} \mid i = 1, \ldots, m\}$. The initial predicted permutation starts with these jobs, i.e., $j_1, j_1', \ldots, j_m, j_m'$ and the rest of the jobs follow in an arbitrary order. Once the lengths of the jobs in $P$ are determined, the predictor calculates $\mu(h, P)$ for each $h \in \mathcal{H}$. Then, it makes a switch to its final prediction by ordering the jobs not contained in $P$ according to the hypothesis with the smallest $\mu(h, P)$.

---

**Predictor 12:** non-clairvoyant scheduling, agnostic case

---

1  At time 0: sample $m$ pairs of jobs $P$
2  When the last job in $P$ is completed:
3      **for** $h \in \mathcal{H}$ **do** compute $\mu(h, P)$
4      Switch to a new prediction based on $\hat{h}$ with minimum $\mu(\hat{h}, P)$

---

Predictor 12 makes only one switch which happens at the moment when the last job from $P$ is completed.

**Lemma 27.** *Let* $\pi$ *be the final prediction produced by Predictor 12 during its only switch. With probability* $(1-\delta)$, *we have* $\mu(\pi) \le \mu(h^*) + O(n^{5/3}(\log\frac{\ell}{\delta})^{1/3})$, *where* $h^*$ *denotes the best hypothesis in* $\mathcal{H}$.

*Proof.* The total weight of the mistakes in the last prediction can be bounded by

$$\mu(\pi) \le \mu(\hat{h}) + 2mn, \tag{1}$$

where $\hat{h}$ is the hypothesis chosen at line 4. This follows because the $2m$ jobs belonging to $P$ which are at the beginning of $\pi$ have length at most 1 and each of them delays the completion of at most $n$ jobs. We need to show that, with high probability, we have to show that $\mu(\hat{h})$ similar to $\mu(h^*)$.

For each hypothesis $h \in \mathcal{H}$ and a pair $\{i,j\} \in \binom{[n]}{2}$ chosen uniformly at random, we denote $\rho_h = \mathbb{E}[\mu(h, \{i,j\})] = \mu(h)/\binom{n}{2}$. Since the pairs in $P$ are chosen uniformly at random, we have $\mathbb{E}[\mu(h, P)] = \rho_h m$ for each $h \in \mathcal{H}$. Now we use Hoeffding's concentration inequality [40, Thm 2.2.6] to show that $\mu(h, P)$ is close to its expectation with a high probability. Choosing $m = \log(2\ell/\delta)/2\epsilon^2$, where $\epsilon$ is a parameter to be decided later, we have

$$\mathbb{P}\big(|\mu(h, P) - \rho_h m| > \epsilon m\big) \le 2\exp\big(-2(\epsilon m)^2/m\big) = 2\exp(-2\epsilon^2 m) \le \delta/\ell,$$

for each $h \in \mathcal{H}$. So, by a union bound, with probability at least $1-\delta$, we have $|\mu(h, P) - \rho_h m| \le \epsilon m$ for all hypothesis $h \in \mathcal{H}$ and the chosen hypothesis $\hat{h}$ must have $\rho_{\hat{h}} \le \rho_{h^\star} + 2\epsilon$. Multiplying by $\binom{n}{2}$ we get $\mu(\hat{h}) \le \mu(h^*) + 2\epsilon\binom{n}{2}$ with probability $1 - \delta$.

The total weight of mistakes in the final prediction $\pi$ is bounded as follows:

$$\mu(\pi) \le \mu(h^*) + 2\epsilon\binom{n}{2} + \frac{\log(2\ell/\delta)}{\epsilon^2}n.$$

We choose $\epsilon = O\big(\frac{\log(\ell/\delta)}{n}\big)^{1/3}$, getting $\mu(\pi) \le \mu(h^*) + O\big(n^{5/3}(\log\frac{\ell}{\delta})^{1/3}\big)$ with desired probability. $\square$

**Algorithm.** At each step, it chooses the first unfinished job in the predicted ordering and runs it until it is completed.

**Lemma 28.** *Given a predictor which makes switches only at moments when some job is completed, never changes ordering of finished jobs, and the total weight of the mistakes in its final predictions is* $\mu$, *the regret of Algorithm 13 is* $\mu$.

---

**Algorithm 13:** non-clairvoyant scheduling, agnostic case

---

**1 for** *each time instant $t$* **do**

**2** $\quad$ run the first unfinished job according to the current prediction $\pi$

---

*Proof.* If switches happen only at job completions then Algorithm 13 never preempts any job before it is finished. Since, the predictor changes only the ordering of the unfinished jobs during, the algorithm processes jobs in the order suggested by the final prediction. By Proposition 26, the difference between the cost incurred by the algorithm and the offline optimum is equal to $\mu$. $\qquad\square$

Algorithm 13 when run with Predictor 12 first processes jobs in $P$. Once the last job in $P$ is completed, Predictor 12 switches to its final prediction by updating the ordering of unfinished jobs, fulfilling the conditions of Lemma 28. Together with Lemma 27, we get the following theorem.

**Theorem 29.** *Consider an instance $I$ with maximum job length $1$. There is an algorithm for the agnostic setting which, given a hypothesis class $\mathcal{H}$ of size $\ell$ with error $\mu$, produces a schedule with total completion time at most $\mathrm{OPT}(I) + \mu + O(n^{5/3}(\log \frac{\ell}{\delta})^{1/3})$ with probability at least $(1 - \delta)$.*

## C.3  Lower Bound

In this section, we prove a lower bound for instances with three distinct job lengths. The instances used in our construction will use only integer job lengths and the following technical lemma helps simplifying the exposition of the lower bound.

**Lemma 30.** *Consider instance of non-clairvoyant scheduling with jobs of integer lengths. Any online algorithm $A$ on this instance can be converted to an online algorithm $A'$ with no larger cost which interrupts and starts processing of jobs only at integer time steps.*

*Proof.* Let $t_1, \ldots, t_N$ be time instants such that the processed part of some job in the schedule produced by $A$ reaches an integer value. We use the following notation: The *milestone $i$* reached at time $t_i$ is described by $k_i \in \mathbb{N}$ and $j_i \in \{1, \ldots, n\}$ meaning that $k_i$ units of job $j_i$ become completed at $t_i$. Since jobs are guaranteed to have integral lengths, algorithm $A$ discovers new information about job lengths only at times $t_1, \ldots, t_N$. Namely, at time $t$, it knows that the length of a job $j$ is $\max\{k_i \mid t_i \leq t \text{ and } j_i = j\}$ if $j$ is finished, and at least $\max\{k_i + 1 \mid t_i \leq t \text{ and } j_i = j\}$ if $j$ is unfinished.

We describe algorithm $A'$ which reaches the milestones $1, \ldots, N$ in the same order as $A$, reaching milestone $i$ at time $i \leq t_i$. At time $t = 0$, it chooses job $j_1$ and processes it for a single time unit, reaching milestone $1$ with $j_1$ and $k_1 = 1$ at time $1 \leq t_1$. Having $i$ milestones reached at time step $i \in 1, \ldots, N - 1$, $A'$ chooses job $j_{i+1}$ and processes it for a single time unit. Since the previous milestone involving $j_{i+1}$ was already reached by $A'$, $j_{i+1}$ is processed for $k_{i+1}$ time units, reaching milestone $i + 1$. We have $t_{i+1} \geq i + 1$, because reaching each milestone requires a single unit of computational time and no algorithm can reach $i + 1$ milestones before time $i + 1$.

Since all jobs have integer lengths, they can be completed only when some milestone is reached. Since $A'$ reaches all milestones no later than $A$, its total completion time is at most the one achieved by $A$. $\qquad\square$

**Lemma 31.** *Consider two schedules such that the second one differs from the first one by moving at least a single unit of some job $j$ earlier, jumping over completion times of $d$ jobs, while the completion time of $j$ does not change. Then the total completion time of the second schedule is larger by at least $d$.*

*Consider two schedules such that the second one differs from the first one by moving a whole job $j$ of size $3$ earlier, jumping over $d$ jobs of size at most $2$. Then the total completion time of the second schedule is larger by at least $d$.*

*Proof.* First case: the completion times of $d$ delayed jobs increase by $1$ and all other completion times remain the same.

Second case: the completion time of $j$ decreases by at most $2d$, while the completion times of the delayed jobs increase by 3. All other completion times do not change. □

**Theorem 32.** *There is a hypothesis class of size $\ell$ such that no algorithm for non-clairvoyant scheduling in realizable setting can achieve a regret bound $o(\ell n)$.*

*Proof.* We construct $\ell$ instances with $n \geq 2\ell^2$ jobs. The job lengths will be $1, 2, 3$. However, we can rescale the time to make an execution of a job of size 3 to last 1 time unit. Such rescaling changes both the optimal and algorithm's total completion time as well as their difference by factor of 3.

The first $\ell^2$ jobs are divided into $\ell$ blocks of $\ell$ jobs each. The $i$th block includes jobs $(i-1)\ell+1, (i-1)\ell+2, \ldots, i\ell$. The jobs in the $i$th block have length 1 in instance $i$ and 3 in all other instances. The jobs $\ell^2 + 1, \ldots, n$ have length 2 in all instances.

The correct instance is picked uniformly at random. By Yao's principle (see, e.g., [11]), it is enough to prove a lower bound for any deterministic algorithm on this randomized instance to get a lower bound for any randomized algorithm. The optimal solution of any of these instances is to schedule $\ell$ jobs of size 1 first, then $n - \ell^2$ jobs of size 2, and finally $\ell^2 - \ell$ jobs of size 3.

We assume that the algorithm starts and preempts jobs only in integral time steps. This assumption is without loss of generality by Lemma 30. When deciding which job to run at time $t \in \mathbb{N}$, it can choose either a job from $1, \ldots \ell^2$ or a job from $\ell^2 + 1, \ldots n$. If it runs a job from $1, \ldots, \ell^2$ for at least 1 time unit, it discovers the length of all jobs in the same block. If the true size of the job was 1 and the job is finished, it also discovers the true instance. We denote by $A \subseteq \{1, \ldots, \ell\}$ the set of blocks such that the algorithm still does not know the lengths of the jobs in these blocks. We consider the following two cases:

- Algorithm runs a job $j \in \{\ell^2 + 1, \ldots, n\}$: such a job has size 2 in all instances. If the correct instance is not yet determined and there are still $\ell$ unfinished jobs of length 1, this action worsens the algorithm's schedule by $\ell$, by Lemma 31.

- Algorithm runs a job $j \in \{1, \ldots, \ell^2\}$: the length of $j$ is 1 with probability $1/|A|$ if $j$ belongs to a block $i \in A$ and 0 otherwise. If it is 1, the algorithm determines the correct instance and suffers no more regret. Otherwise, $|A|$ decreases by 1. If ALG processes this job completely, it suffers regret $\geq r_t$ by Lemma 31, where $r_t$ is the number of unfinished jobs of size 2. If it processes the job for at least 1 time unit without finishing it, it also suffers regret $\geq r_t$.

One of the following complementary events occurs with probability at least 1/2: When the first job of size 1 is scheduled, either (1) $r_t < (n - \ell^2)/2$ or (2) $r_t \geq (n - \ell^2)/2$.

If the event (1) occurs, at least $(n - \ell^2)/2$ jobs of size 2 are scheduled before the first job of size 1, each of them causes regret at least $\ell$. With $n \geq 2\ell^2$, we have the regret at least $\Omega(\ell n)$.

In case event (2) occurs, we calculate how many jobs of size 3 are scheduled before the first job of size 1 in expectation. If each job $j \in \{1, \ldots, \ell^2\}$ chosen by the algorithm belongs to a different block, it will run $\frac{\ell^2+1}{\ell+1} \geq \ell/2$ jobs of size 3 before it runs the first job of size 1 in expectation. Otherwise, the expectation is even higher. Therefore, the algorithm suffers regret at least $(\ell/2)r_t \geq (\ell/2)(n - \ell^2)/2 \geq (\ell/2)(n/4) = \Omega(\ell n)$. □

## D   Caching in Agnostic Setting

We describe extensions of our results from Section 4 to agnostic setting and setting with a more natural hypotheses. We also prove our lower bounds.

In agnostic setting, we are given a set of $\ell$ hypotheses $\mathcal{H} = \{r^1, \ldots, r^\ell\}$ which does not necessarily contains the input sequence $r$. The number of mistakes of hypothesis $r^i$ is defined as the number of time steps $t \in \{1, \ldots, T\}$ such that $r_t^i \neq r_t$. The best hypothesis is the one with the smallest number of mistakes.

**Predictor:** At each time step $t$, the predictor chooses a hypothesis $i$ at random according to the probability distribution produced by the HEDGE algorithm [31, 18]. If $i$ is different from the hypothesis chosen at time $t - 1$, there is a switch to a new prediction $\pi$ which is consistent with the previous prediction until time $t$ and, for $\tau = t, \ldots, T$, the predictor updates $\pi_\tau := r_\tau^i$.

We construct a sequence of loss functions as an input to HEDGE: At time $t$, the loss of hypothesis $i$ is 1 if it predicts an incorrect request and 0 otherwise. At each time step $t$, HEDGE gives us a probability distribution $\xi^t$ over the hypotheses. The predictor samples a hypothesis from this distribution in the following way. Let $f$ be a min-cost flow of the probability mass from distribution $\xi^{t-1}$ to distribution $\xi^t$ (i.e., $\sum_{j=1}^\ell f_{ij} = \xi_i^{t-1}$ and $\sum_{i=1}^\ell f_{ij} = \xi_j^t$), where flow $f_{ij}$ has cost 1 if $i \neq j$ and 0 otherwise.[8] Note that the cost of this flow $\sum_{i \neq j} f_{ij}$ is equal to Total Variation Distance (TVD) between $\xi^{t-1}$ and $\xi^t$. If hypothesis $i$ was chosen at time $t - 1$ as a sample from $\xi^{t-1}$, then the predictor switches to a hypothesis $j$ with probability $f_{ij}/\xi_i^{t-1}$. This way, the probability of choosing $j$ at time $t$ is

$$\sum_{i=1}^\ell \mathbb{P}(i \text{ chosen at } t-1)\mathbb{P}(\text{switching to } j \mid i \text{ chosen at } t-1) = \sum_{i=1}^\ell \xi_i^{t-1} \frac{f_{ij}}{\xi_i^{t-1}} = \xi_j^t$$

and the probability of a switch occurring is $\sum_{i \neq j} f_{ij} = \text{TVD}(\xi^{t-1}, \xi^t)$. See Predictor 14 for a summary.

---

**Predictor 14:** caching, agnostic setting

---

1   $\eta := \ln \frac{1}{1-1/k}$ ;                      `// Learning rate parameter`

2   $w := (1, \ldots, 1)$, $\xi^0 := w/\ell$;

3   predict a hypothesis sampled from $\xi^0$;

4   **for** $t = 1, \ldots, T$ **do**

5      **for** $i \in \{1, \ldots, \ell\}$ s.t. $r_t^i \neq r_t$ **do**

6         $w_i := w_i \exp(\eta)$ ;     `// loss of instance i is 1:  update w_i using HEDGE`

7      $\xi_i^t := w_i / \sum_{i=1}^\ell w_i$ for each $i = 1, \ldots, \ell$ ;        `// Update distribution over hypotheses`

8      compute min-cost flow $f$ from $\xi^{t-1}$ to $\xi^t$, so that $\sum_{i=1}^\ell f_{ij} = \xi_j^t$;

9      if the previous prediction was hypothesis $i$, switch to $j$ w.p. $f_{ij}/\xi_i^{t-1}$;

---

The following performance bound of the HEDGE algorithm can be found, e.g., in [12, Thm 2.4].

**Proposition 33.** *Let $\mu^\star$ be the loss of the best of the hypotheses. Then, the expected loss of the HEDGE algorithm with learning rate parameter $\eta$ is*

$$\mu \leq \frac{\eta\mu^\star + \ln\ell}{1 - \exp(-\eta)}.$$

The probability distribution produced by HEDGE is relatively stable. We can bound $\text{TVD}(\xi^{t-1}, \xi^t)$ as a function of $\eta$ and the expected loss incurred by HEDGE at time $t$.

**Proposition 34** (Blum and Burch [10] (Thm 3)). *Let $\xi^{t-1}$ and $\xi^t$ be probability distributions of HEDGE learning rate parameter $\eta$ at times $t - 1$ and $t$ respectively. Then, we have*

$$\text{TVD}(\xi^{t-1}, \xi^t) \leq \eta \sum_{i, r_t^i \neq r_t} \xi_i^t.$$

**Lemma 35.** *Let $\mu^\star$ be the number of mistakes of the best hypothesis. The predictor for the agnostic setting makes $\mu$ mistakes and $\sigma$ switches in expectation, where*

$$\mu \leq (1 + 1/k)\mu^\star + k\ln\ell \quad and \quad \sigma \leq (1/k + 1/k^2)\mu.$$

---

[8]In our situation, the min-cost flow $f_{ij}$ can be expressed using an explicit formula: we define $\delta = \xi^t - \xi^{t-1}$ and write $f_{ij} = (-\delta_i)^+ \frac{\delta_j^+}{\sum_{m=1}^\ell \delta_m^+}$ for each $i, j \in \{1, \ldots, \ell\}$, using notation $x^+ := \max\{0, x\}$.

*Proof.* The number of mistakes made by Predictor 14 is equal to the loss achieved by HEDGE algorithm. By Proposition 33, we have

$$\mu \leq \frac{\eta\mu^\star + \ln \ell}{1 - \exp(-\eta)}. \tag{2}$$

We choose $\eta := \ln \frac{1}{1-1/k}$ which is $\leq 1/k + 1/k^2$, whenever $k \geq 4$. This is to make $k\sigma = k\eta\mu$ comparable to $\mu$. We substitute this upper bound in 2, and get

$$\mu \leq \frac{(\ln \frac{1}{1-1/k})\mu^\star + \ln \ell}{1/k} \leq (1 + 1/k)\mu^\star + k\ln \ell.$$

At each time step $t$, Predictor 14 makes a switch with probability $\sum_{i \neq j} f_{ij} = \text{TVD}(\xi^{t-1}, \xi^t)$. By Proposition 34, the expected number of switches made by our predictor is

$$\sigma \leq \sum_{t=1}^{T} \eta \sum_{i, r_t^i \neq r_t} \xi_i^t = \eta\mu \leq (1/k + 1/k^2)\mu,$$

since $\sum_{i, r_t^i \neq r_t} \xi_i^t$ is the expected number of mistakes at time $t$. $\qquad\square$

**Algorithm.**    Our algorithm follows the FitF solution recomputed at each switch. If it happens that the current request $r_t$ is not served by this solution (i.e., there is a mistake in the prediction), the algorithm loads $r_t$ ad-hoc and removes it instantly in order to return to the FitF solution.

---

**Algorithm 15:** caching, agnostic setting

---
1  **for** $t = 1, \ldots, T$ **do**
2      **if** $t = 1$ *or there is a switch* **then**
3          recompute $x_1, \ldots, x_T$ the FitF solution for the current prediction;
4      move to $x_t$;
5      **if** $r_t \notin x_t$ **then**
6          load $r_t$, evicting an arbitrary page, and serve $r_t$;
7          return back to $x_t$;

---

If the predictor makes 0 mistakes, i.e., $\mu = 0$, then $r_t \notin x_t$ never happens and this algorithm is the same as the one for the realizable case (but note that the predictor is different). If $\mu > 0$, then it suffers an additional cost of 2 for every mistake.

**Lemma 36.** *Consider an input sequence $r$ and let $\text{OPT}(r)$ be the cost of the optimal offline solution for this sequence. If the predictor makes $\mu$ mistakes and $\sigma$ switches during this sequence, then the algorithm above has cost at most*

$$\text{OPT}(r) + 4\mu + k\sigma.$$

*Proof.* For $t = 1, \ldots, T$, let $\pi_t$ denote the prediction made for $r_t$ at time $t$. Here, $\pi_t = r_t$ if the predictor was right or decided to make a switch. Otherwise, the predictor chose to suffer a mistake.

If the real input was $\pi_1, \ldots, \pi_T$, the cost of the algorithm would be at most $\text{OPT}(\pi) + k\ell$ by Lemma 6. Since $\pi$ and $r$ differ in $\mu$ time steps, the cost of the algorithm is at most $\text{OPT}(\pi) + k\sigma + 2\mu$.

Now, note that we can use the optimal solution $x_1, \ldots, x_T$ for $r$ to produce a solution for $\pi$ of cost at most $\text{OPT}(r) + 2\mu$: whenever $\pi_t \notin x_t$ (this can happen only if $\pi_t \neq r_t$), we evict an arbitrary page to load $\pi_t$, serve the request and return back to $x_t$. This implies $\text{OPT}(\pi) \leq \text{OPT}(r) + 2\mu$.

Therefore, the cost of our algorithm is at most

$$\text{OPT}(r) + 4\mu + k\sigma. \qquad\square$$

**Theorem 37.** *With a class $\mathcal{H}$ of $\ell$ hypothesis, Algorithm 15 has expected cost at most*

$$\text{OPT} + (5 + 6/k)\mu^\star + (2k + 1)\ln \ell$$

*where $\mu^\star$ denotes the minimum number of mistakes over hypotheses in $\mathcal{H}$ and $\text{OPT}$ the cost of the offline optimum.*

*Proof.* By Lemma 36 and Lemma 34, the cost of the algorithm is at most

$$\text{OPT}(r) + 4\mu + k\sigma$$
$$\leq \text{OPT}(r) + 4(1 + 1/k)\mu^\star + k\ln\ell + k(1/k + 1/k^2)(1 + 1/k)\mu^\star + k(1/k + 1/k^2)k\ln\ell$$
$$\leq \text{OPT}(r) + (5 + 6/k)\mu^\star + (2k + 1)\ln\ell. \qquad \square$$

## D.1 Achieving Robustness

We can robustify our algorithms both for the realizable and agnostic setting in the following way. We partition the input sequence into intervals such that the cost paid by OPT in interval $i$ is $2^i k$. Note that we start with an empty cache and $\text{OPT} \geq k$ on any non-trivial instance: if $\text{OPT} \leq k$, less than $k$ distinct pages were requested and any lazy algorithm is optimal. In each of these intervals, we first run our algorithm until its cost reaches $2^i k \log k$. Then, we switch to an arbitrary worst-case algorithm with competitive ratio $O(\log k)$ for the rest of the time window.

**Lemma 38.** *Consider an algorithm* ALG *which, given a predictor making $\mu$ mistakes and $\sigma$ switches achieves regret $\alpha\mu + \beta k\sigma$. We can construct an algorithm* ALG$'$ *which is $O(\log k)$-competitive in the worst case and its regret is always bounded by $O(\alpha)\mu + O(\beta)k\sigma$.*

*Proof.* We denote $G$ the set of intervals $i$ where ALG paid cost $\text{ALG}_i \leq 2^i k \log k$. The cost of ALG$'$ is then

$$\text{ALG}' \leq \sum_{i \in G} \text{ALG}_i + \sum_{i \notin G}(O(\log k)\, 2^i k + 2k), \leq \sum_{i \in G} \text{ALG}_i + \sum_{i \notin G} O(\log k)\, 2^i k,$$

where $2k$ denotes the cost of switching to the worst-case algorithm and back to ALG. This already shows that ALG$'$ is $O(\log k)$-competitive: we have $\text{ALG}' \leq O(\log k)\,\text{OPT}$, because OPT pays $2^i k$ in window $i$.

To show that it still preserves good regret bounds, note that $\alpha\mu_i + \beta k\sigma_i \geq 2^i k \log k$ in each interval $i \notin G$, where $\sigma_i$ denotes the number of switches and $\mu_i$ the number of mistakes in interval $i$. Therefore, we have

$$\text{ALG}' \leq \text{ALG} + O(\alpha)\mu + O(\beta)k\sigma. \qquad \square$$

## D.2 Extension to Next-arrival-time Predictions

In order to compute a step of FitF, we either need to know the whole request sequence or, at least, times of next arrivals of the pages in our current cache. Lykouris and Vassilvitskii [32] proposed acquiring a prediction of the next arrival time (NAT) of each page when requested. Precise NAT predictions allow us to simulate FitF. Lykouris and Vassilvitskii [32], Rohatgi [36], Wei [41] proposed algorithms able to use them even if they are not precise.

Our result can be extended to setting where each hypothesis is not an explicit caching instance given in advance but rather a set of prediction models which generate the relevant parts of the instance over time. Consider models generating next-arrival-time predictions, as considered by Lykouris and Vassilvitskii [32]. Given the part of the sequence seen so far, they produce a next arrival time of the currently requested page. Using this information, we can compute the FitF solution to an instance which fully agrees with the predictions. Moreover, we can easily detect mistakes by comparing the currently requested page with the page which was predicted to arrive at the current moment. Therefore, Theorem 7 and Theorem 37 hold also with $\mathcal{H}$ containing $\ell$ models producing next-arrival-time predictions.

## D.3 Lower Bounds

**Lemma 39.** *In realizable setting, there is an input instance and a class $\mathcal{H}$ of $\ell$ hypotheses such that any (randomized) algorithm has regret at least $\frac{k}{2}\log\ell$.*

*Proof.* Let $\ell$ be power of two. We use universe of pages $U = \{1, \ldots, 2k\}$ and define sequences $a = 1, \ldots, k$ and $b = k+1, \ldots, 2k$. By concatenating $a$ and $b$ in a specific manner, we construct building blocks of the input sequence and the hypotheses: $\sigma_0 = a^k b a^{2k+1}$ and $\sigma_1 = a^k b b^k b a^k$. Here $a^k$ denotes a sequence $a$ iterated $k$ times, and both blocks are chosen to have equal length.

For each $i = 1, \ldots, \ell$, we use its binary representation $b_1^i b_2^i \cdots b_{\log \ell}^i$ to construct a hypothesis $r^i = \sigma_{b_1^i} \sigma_{b_2^i} \cdots \sigma_{b_{\log \ell}^i}$ from blocks $\sigma_0$ and $\sigma_1$. Note that for any input sequence constructed from blocks $\sigma_0$ and $\sigma_1$, we can construct an offline solution such that:

- at the beginning and at the end of each block, its cache content is $\{1, \ldots, k\}$.

- during block $\sigma_0$, it keeps pages $1, \ldots, k - 1$ in cache, paying $k$ for page faults during sequence $b$ and $1$ for loading page $k$, i.e., $k + 1$ page faults in total

- during block $\sigma_1$, it pays $2k$, because it replaces the whole cache with $\{k+1, \ldots, 2k\}$ during the first occurrence of $b$ and then with $\{1, \ldots, k\}$ after the last occurrence of $b$.

For each $i = 1, \ldots, \log \ell$, we issue $a^k b$ – the common prefix of $\sigma_0$ and $\sigma_1$ – and compute $n_i$ the expected number of pages from $\{1, \ldots, k\}$ in the cache. Note that any algorithm has to pay at least $k$ during $a^k b$ since it contains $2k$ distinct pages. If $n_i < k/2$, we issue $a^{2k+1}$, completing block $\sigma_0$. Otherwise, we issue $b^k b a^k$, completing block $\sigma_1$.

In the first case, its expected cost will be at least $k + (k - n_i) > k + k/2$, because the algorithm will have at least $k - n_i$ page faults during the sequence $a^{2k+1}$ at the end of $\sigma_0$.

In the second case, the expected cost of the algorithm will be at least $k + n_i + k \geq 2k + k/2$, where $k$ page faults are during $a^k b$, $n_i$ page faults during $b^k$, and another $k$ page faults during for $b a^k$ (contains $2k$ distinct pages). In both cases, the difference from the cost of the offline solution is at least $k/2$.

This way, we constructed an instance $j \in \{1, \ldots, \ell\}$ with binary representation $b_1 b_2 \cdots b_{\log \ell}$, where $b_i = 0$ if $n_i < k/2$ and $b_i = 1$ otherwise. Moreover, in each iteration $i = 1, \ldots, \log \ell$, the algorithm pays at least by $k/2$ more compared to the offline solution. □

**Lemma 40.** *There is no deterministic algorithm which, given a predicted request sequence with $\mu$ mistakes achieves regret smaller than $\mu$.*

*Proof.* We construct a predicted instance $\pi = ((1, \ldots, k, 0)(2, \ldots, k)(0, 1, \ldots, k))^n$ which is given to the algorithm ALG. We construct a real instance which is constructed online based on ALG's decisions.

For any iteration $i = 1, \ldots, n$, we build a corresponding part of the real instance which differs from the predicted one in at most one request and show that ALG has one more page fault compared to a described adversarial strategy ADV which starts and ends each iteration with cache content $\{1, \ldots, k\}$.

First, any algorithm has a page fault during $(1, \ldots, k, 0)$ because $k + 1$ distinct pages are requested, so both ALG and ADV pay 1. At the moment when 0 is requested, ALG must be missing some page from $p \in \{1, \ldots, k\}$. If $p = 1$, ADV evicts $k$ instead and the real request sequence continues with $(2, \ldots, k - 1, 1)$ instead of $(2, \ldots, k)$, causing a single mistake in the prediction. ALG has a page fault during this part, while ADV has no page fault. If $p \in \{2, \ldots, k\}$, ADV evicts 1 and the real request sequence continues as predicted without any mistake, causing a page fault to ALG, while ADV has no page fault. In the last part $(0, 1, \ldots, k)$, both ALG and ADV have a page fault. So, ALG had at least 3 page faults while ADV only 2, and there was at most one mistake.

Therefore, the total cost of ALG is at least $3n$ while ADV pays $2n$. Since $\mu \leq n$, the regret of ALG is at least $\mu$. □

