# OpenReview forum: "Learning-Augmented Algorithms with Explicit Predictors"
_NeurIPS.cc/2024/Conference — NeurIPS 2024 poster_

### Official Review · Reviewer_oKpi · 2024-07-08

**Soundness:** 3
**Presentation:** 3
**Contribution:** 3
**Rating:** 6
**Confidence:** 4

**Summary:**

The paper introduces a new framework for learning-augmented online algorithms. Learning-augmented algorithms (aka algorithms with predictions) are a very active subfield of beyond worst-case algorithm analysis. For online algorithms, which cope with uncertainty on their input, it gives an algorithm an addition information in form of a prediction. This prediction can be the (unknown) online instance or any other form of additional information. A learning-augmented algorithm is usually analyzed w.r.t. the quality of the given prediction.
In the most commonly used setup for online algorithm, the prediction is assumed to be generated upfront from some black-box predictor, or arrives online together with requests. There are also models where multiple predictions are initially available to an algorithm.
The authors of this paper introduce a new framework which integrates both the predictor and the learning-augmented algorithm.
The main motivation is that a predictor might update its prediction while the online instance is being revealed, and, thus, increase the prediction's quality over time.
In their model, they assume that there are given a set of hypotheses, which can be thought of different characteristics which the coming input could have.
The predictor can compute at any time a new prediction using these hypotheses and the input which has been revealed up to that time.
The authors distinguish between two settings: the easier "realizable" setting, which assumes that the hypothesis which corresponds to the actual input is contained in the hypotheses class, and the harder "agnostic" setting, which does not make this assumption.
The authors then apply this framework to three well-studied online problems in the learning-augmented area: caching, load balancing and non-clairvoyant scheduling. Some of their new bounds improve over previous work.

**Strengths:**

- The paper introduces an interesting generalization of the traditional learning-augmented framework. I think that this new view on learning-augmented algorithm can have an impact on this rapidly evolving field, since it untangles concepts which have already been used partially. I also appreciate the clear structure of the framework into predictor and algorithm.
- The framework is conceptually well motivated and the authors show that it can be applied to at least three different and well-studied problems, which proves that it is practicable and can be used for relevant problems.
- The authors present improved guarantees for previously well-studied online problems.

**Weaknesses:**

- A minor weakness can be the algorithmic novelty for the concrete applications. It seems that the algorithms follow in many cases the predictions given by the predictor. The predictors are tailored to the specific applications and are also rather simple.
- Another minor weakness is the close relation to the established 'multiple prediction' setting. From my understanding, one can see the hypothesis class the set of given predictions, and the goal (in the agnostic setting) is to guarantee a bound w.r.t. the best one.

**Questions:**

- As far as I understand it one can model the traditional black-box setting via the agnostic setting by using a singleton hypothesis class and the 'traditional' prediction error as loss. Is this true? I am not sure if I missed this, but I think this could also be an interesting insight.


Further comments:
- Line 1140: there is a comma before the second inequality.

**Limitations:**

As far as I see, all limitations have been properly addressed.

---

> ### Author Rebuttal · Authors · 2024-08-06
>
> We thank the reviewer for highlighting the positive aspects of our work and pointing out a typo.
>
> > As far as I understand it one can model the traditional black-box setting via the agnostic setting by using a singleton hypothesis class and the 'traditional' prediction error as loss. Is this true?
>
> Yes, the classical setting of learning-augmented algorithms is indeed subsumed by the case where the hypothesis class contains only a single hypothesis. We will make sure to clarify this in the final version of our manuscript.
>
> > A minor weakness can be the algorithmic novelty for the concrete applications.
>
> In our framework, we achieve better bounds than the works on multiple black-box predictions using simpler algorithms (e.g. see our results for non-clairvoyant scheduling and caching).
> This can be viewed as a strength rather than a weakness. Moreover, the idea of separating the learning and algorithmic parts is novel.
>
> > Another minor weakness is the close relation to the established 'multiple prediction' setting.
>
> Our setting is able to model the established 'multiple prediction' setting. However, our setting is more general and flexible. As can be seen from our results, it naturally leads to clean and simple algorithms which we find important.

---

> > ### Comment · Reviewer_oKpi · 2024-08-10
> > **Response to Rebuttal**
> >
> > I thank the authors for their rebuttal, and for answering my question about the connection to the traditional setting.
> >
> > My opinion about the paper and my score remain unchanged.

---

### Official Review · Reviewer_1P4N · 2024-07-09

**Soundness:** 4
**Presentation:** 3
**Contribution:** 3
**Rating:** 6
**Confidence:** 5

**Summary:**

This paper proposes a new framework to use machine learned predictions to improve online algorithms. Recent work that has done this (in a slightly different framework) goes under the name learning-augmented algorithms (LA algorithms) or algorithms with predictions. The main difference in this work compared to the usual LA algorithms is that these authors propose integrating the learning problem into the algorithmic problem by allowing the algorithm to adaptively update its prediction based on the new information it receives. Additionally, instead of maintaining a single prediction, this framework maintains a hypothesis class of information about the instance. The paper exemplifies this framework on caching, online load balancing, and online non-clairvoyant scheduling. For all 3 problems they study a realizable and agnostic setting, where in the realizable setting, the hypothesis class contains h(I) for I the actual underlying offline instance, and in the agnostic setting, the information corresponding to the true offline instance is not necessarily in the class.

**Strengths:**

- This framework could indeed be a better way to study algorithms with predictions. In combining the learning process with the algorithmic procedure we get the benefit of (1) demystifying how the prediction is obtained and (2) allowing ourselves the flexibility of updating the prediction as we receive new information about the input.
- Using this more all-encompassing approach, the framework is able to improve upon some bounds from LA algorithms.
- The authors did a good job in the related work differentiating their model from that on algorithms with a prediction pf portfolios, as well as data driven algorithms.

**Weaknesses:**

The algorithms proposed are augmenting simple procedures with this richer, more flexible hypothesis class, and the analyses involved are very straightforward. This isn’t inherently a bad thing. But there is a lot of work in the LA algorithms scene lately, and the work that stands out to me the most is the work that highlights some interesting technicality of the problem that wasn’t understood through classic worst-case analysis approaches. I’m afraid the problems that this model has been applied on did not result in technical components that stood out, and thus could lead to this paper blending into an already noisy scene. I will leave more specific related questions that may address this fear under “questions”.

**Questions:**

- How do you see this framework allowing us to understand the complexity of problems (from a beyond-worst case analysis perspective) in a better way than the standard LA algorithms framework or other beyond worst-case analysis frameworks? In other words, how does the ability to (1) have a hypothesis class of information for the prediction and (2) update the prediction allow us to further surpass worst case lower bounds?
- It’s unclear to me how large the hypothesis class should be. You get this trade off in the agnostic setting between having a large \ell should lead to smaller \mu^* indoor bounds, or a small \ell might imply larger \mu^*. Have you thought at all for the problems you study about how you can balance these?
- A few times, you mention something like “each hypothesis h(I) provided sufficient information about the instance in order to determine an offline optimal solution OPT(I)”. This statement cannot be literally true, as you study a problem whose offline version is NP hard. I think you mean something like, h(I) provides sufficient information about the instance in order to find an offline solution \alpha OPT(I) for alpha the apx factor, and then alpha comes into your completive ratio or regret bound.
- Something seems weird in the appendix, section B.1 and C.1 have the exact text repeated. It seems B.1 is maybe a totally unnecessary section.

**Limitations:**

yes

---

> ### Author Rebuttal · Authors · 2024-08-06
>
> We thank the reviewer for their feedback. Most (if not all) works on learning augmented algorithms solve some learning problem in an ad-hoc manner by assuming a black-box access to some predicitions. In our framework, we "open the box" and define the learning problem explicitly. This allows for a deeper and methodical integration of results from learning theory in learning-augmented algorithms, and as we demonstrate, gives rise to simple and clean algorithms with better performance bounds than previous works. We hope that our "white-box" perspective will inspire further work.
>
> > how does the ability to (1) have a hypothesis class of information for the prediction and (2) update the prediction allow us to further surpass worst case lower bounds?
>
> We use hypothesis classes to model the learning problem (as is standard in learning theory). Thus, when the input is compatible with one of the hypothesis in the class, our algorithms can outperform algorithms in the standard learning augmented framework.  This can be illustrated by a simple example of a bi-modal input: there are only two possible scenarios, and there is no single set of pre-computed predictions which works well for both.  In our framework, we can learn the actual input sequence while processing it and then apply the approriate set of predicitons. As illustrated in the paper, this idea naturally extends to more than just two inputs and as well as to noisy scenarios.
>
> Similar situations happen in other frameworks of beyond-worst case analysis. For example, the input may be stochastic in the first scenario
> while possessing some specific pattern in the second scenario.
>
> One might argue that the argument above does not apply to problems like caching, where previous works did consider predictions that are generated over time. As noted by reviewer qEgX, in such a setting, the predictor may adapt its predictions based on the part of the input seen so far. Our framework captures this in a principled way which allows
> theoretical analysis of such adaptability without committing to specific ad-hoc predictions.
>
> > It’s unclear to me how large the hypothesis class should be.
>
> In principle, the hypothesis class reflects prior knowledge the algorithm design has on the learning problem. In practice the class can be determined e.g. using collected statistics of past inputs.
> Our framework is effective when one can indeed define a restricted hypothesis class which can capture the regularities in the inputs. (Otherwise we are back in the classical worst-case setting where any input is possible.)
>
> Our framework allows one to try and reduce the size of the hypothesis class by ``clustering'' the hypotheses. Indeed this is likely to increase the distance of the actual input to the class.  We specify the performance as a function of the size of the class and the maximum distance of an input to the class. Studying the relation between these parameters for specific applications is a nice direction for further research.
>
> > A few times, you mention something like “each hypothesis h(I) provided sufficient information about the instance in order to determine an offline optimal solution OPT(I)”. This statement cannot be literally true, as you study a problem whose offline version is NP hard.
>
> We mean that $h(I)$ contains enough information to determine an offline optimal solution. We do not require that this could be done in polynomial time nor use it to compute an optimal solution: e.g. our algorithm for load balancing uses $h(I)$ to compute an approximate solution in polynomial time.
>
> Thank you for pointing out that B.1 and C.1 are repeated. We will fix this.

---

> > ### Comment · Reviewer_1P4N · 2024-08-11
> >
> > I thank the authors for their response.
> >
> > The ability to combine the learning problem with the algorithmic one is obviously desirable. I think where my initial criticism came from is that I'm not totally sure the paper's proposed way of doing this is "the right way". On further thought, this is an unrealistic expectation to have. It may take several tries for researchers on LA algorithms to build consensus on the right model to open up the black box of the learning problem. I'm not sure this is the right model, but it's a good start, and it shows promise given the results they obtain.  From this lens, I think the paper is making an important contribution, and I will be raising my score. I support acceptance.

---

### Official Review · Reviewer_B3ME · 2024-07-12

**Soundness:** 3
**Presentation:** 2
**Contribution:** 4
**Rating:** 7
**Confidence:** 4

**Summary:**

This paper considers a new formulation for learning-augmented algorithms/algorithms with predictions and applies it to the fundamental problems of online caching, online load balancing, and non-clairvoyant scheduling.  In the new formulation, the predictions/predictor is made more explicit and is a part of the solution process.  The predictions are given by a hypothesis class ${\cal H}$ of size $\ell$, which contains information about the "plausible instances" we are trying to solve.  One can think of ${\cal H}$ as coming from past data and the goal is to use this information to get improved algorithms.  This is done in two settings: the realizable and agnostic settings.

In the realizable setting, the true instance is realized in ${\cal H}$, so we can compare to the optimal cost in hindsight.  For this, the paper provides algorithms with the following bounds:

 - Caching: ${\rm OPT}+ k \log \ell$
 - Load Balancing: $O(\log \ell \cdot {\rm OPT})$
 - Non-clairvoyant scheduling: ${\rm OPT} + \ell \sqrt{ 2 \rm OPT}$

In the agnostic setting, the true instance may not be realized in ${\cal H}$, so instead we compare to the best hypothesis in ${\cal H}$.  For this, the paper provides algorithms with the following bounds:

 - Caching: ${\rm OPT} + O(\mu^* + k \log \ell)$
 - Load Balancing: $O( \mu^* \cdot \log \ell \cdot {\rm OPT})$
 - Non-clairvoyant scheduling: ${\rm OPT} + \mu^* +  O(n^{5/3} \log \ell)$

where in each, $\mu^*$ is a problem-dependent notion of distance from ${\cal H}$ to the true instance.

Some lower bounds are given showing tightness or almost-tightness for these results.

**Strengths:**

This paper gives a very different approach to learning-augmented algorithms that is interesting and has the potential to generate more ideas.  While the algorithms once given the predictions are simple (they are all essentially some notion of follow-the-prediction), this work shows that these simple algorithms can work well if fed carefully constructed predictions (given by the prediction algorithms designed in this paper).  Lower bounds are given showing that the results are somewhat tight.

**Weaknesses:**

- I think it would help the presentation to give examples of hypothesis classes to make the main conceptual idea more clear and provide a stronger connection to practice.

- There is no experimental evaluation.  While some works in this area lack an experimental evaluation, there is a standard setup that has been used for the caching problem by Lykouris and Vassilvitskii [32] as well as Antoniadis et al. [4].  Providing such an evaluation could elucidate the construction of hypothesis classes and provide further evidence in favor of this paper's approach.

- The writing has some awkward phrases and grammar throughout, e.g. the following (among others):

   - line 62 "... will be much smaller then ..." -> "... will be much smaller than ..."
   - lines 107-108 "... using arguably simpler approach ..." -> "... using an arguably simpler approach ..."
   - lines 306-307 "... under certain assumptions about input." -> "... under certain assumptions about the input."
   - lines 374-375 "In offline setting, ..." -> "In the offline setting, ..."

**Questions:**

- To what extent can these techniques be generalized?  E.g., what can be said about $k$-server?

 - Only the case of finite ${\cal H}$ is considered.  Can anything be said about infinite but "low complexity" ${\cal H}$?

**Limitations:**

Limitations and assumptions have been made clear.  Potential negative societal impacts from this work are very unlikely.

---

> ### Author Rebuttal · Authors · 2024-08-06
>
> We thank the reviewer for their positive words and for their useful suggestions
> to improve our manuscript.
> We respond the questions of the reviewer explicitly.
>
> > To what extent can these techniques be generalized? E.g., what can be said about -server?
>
> Our framework can be extended to $k$-server, even to any Metrical Task System (MTS).
> In the lines of our results on caching, we can obtain the following bounds.
> In the realizable case, where one of the hypothesis is the input sequence itself,
> we can achieve regret $O(D\ln \ell)$, where $D$ is the diameter of the metric space.
> In the agnostic case, we can obtain regret $O(D\mu^* + D\ln \ell)$,
> where $\mu^*$ is the number of mistakes in the best hypothesis, where a mistake
> is considered each time the predicted request (or cost function) is not
> exactly the same as the real one.
> Our paper is already quite dense and adding MTS and k-server requires several additional definitions. We tried to illustrate the framework without making it too hard to read.
>
> > Only the case of finite $H$ is considered. Can anything be said about infinite but "low complexity" $H$?
>
> The case of infinite hypothesis classes with low complexity of some kind is very interesting. This is a great direction for future work.
>
> > There is no experimental evaluation.
>
> Our work is proposing a new theoretical framework for modeling of learning augmented algorithms. We have decided to focus on theoretical analysis. Experimenting with our framework is an interesting direction for future research.

---

> > ### Comment · Reviewer_B3ME · 2024-08-07
> >
> > Thank you for your response, my evaluation is unchanged.

---

### Official Review · Reviewer_qEgX · 2024-07-12

**Soundness:** 3
**Presentation:** 3
**Contribution:** 3
**Rating:** 5
**Confidence:** 3

**Summary:**

The paper studies three online algorithms in the learning augmented model: caching, load balancing, and non-clairvoyant scheduling. The goal seems to be to 'online learning' flavored learning-augmented algorithms. Rather than having a single predictor that synthesizes or predict something about the online input, the authors propose a framework where we have access to a large class of hypothesis such as the set of all past inputs. The goal is to be competitive with respect to the best hypothesis for the online input in hindsight. For example in the caching problem, the hypothesis are a large set of input sequences, and the goal is to get close to OPT in the case that the online sequence is actually one of these inputs in our set of hypothesis. In the cases where the online input is not part of the set of hypothesis, the authors obtain error guarantees depending on the hamming distance of the online input and the closest one in the hypothesis.

For the problems studied, the authors match or improve upon (in some parameter regimes) prior works which can be interpreted to be in their framework.

**Strengths:**

- The motivation of the formulation is natural: having access say to the entire history of past inputs could allow one to directly learn the best predictor from data
- The framework matches the best bounds for load balancing, and improves the caching result.

**Weaknesses:**

- The $n^(5/3)$ additive error for non-clairvoyant scheduling  seems to be a meaningless guarantee, since there are $n$ jobs which are assumed to have maximum job length $1$. So any scheduling guarantees that the jobs will finish in $n$ time.
- The algorithms, for example for caching, must iterate over the entire set of hypothesis every time, which seems quite inefficient.
- I'm also not sure if the formulation exactly captures the motivation of directly learning a good predictor from data. For example in caching, the performance seems to be 'bottle-necked' by a single past instance in the set of hypothesis, in the sense that we can only ever expect to do as good as the 'most similar' input in our set of past inputs. Rather, it would be more natural if one would learn from all of them simultaneously (so for ex 'learn' something from the first half of one past input and the second half of another). This is not the fault of their algorithms (which already achieve close to optimal performance), but rather the formulation itself. This is why I still think the standard prediction framework maybe more natural since a predictor can learn to synthesize information and implicitly combine different aspects of different past inputs seen, rather than relying on a single past input alone.
- There are no experiments, even simple synthetic ones. It would be interesting to see even in synthetic case if the framework can actually be carried out.
- There are no algorithms in the main body and it is hard to judge algorithmic improvements.

**Questions:**

- Can the authors clarify the n^(5/3) issue?

**Limitations:**

No societal consequences.

---

> ### Author Rebuttal · Authors · 2024-08-06
>
> We thank the reviewer for their feedback.
> However, their claim about our performance bound for non-clairvoyant scheduling being meaningless is incorrect. We study non-clairvoyant scheduling with the classical objective of
> minimizing the *sum* of completion times which has a different magnitude than the length of the schedule (or makespan).
> In particular, if all jobs have size 1 and the length of the schedule is $n$,
> as the reviewer suggests, then the sum of completion times is about $n^2$.
> Our regret bound is of order $n^{5/3}$, i.e. it is sublinear.
>
> > The algorithms, for example for caching, must iterate over the entire set of hypothesis every time.
> >
> > It would be more natural if one would learn from all of them [hypotheses] simultaneously (so for ex 'learn' something from the first half of one past input and the second half of another).
>
> Our paper describes a basic framework, and we focused mainly on the approximation guarantees and not on other (important) resources such as running time. However, our framework does allow for optimization of other desired resources. For example, to avoid the need of iterating over all hypotheses in caching, we can replace HEDGE in our predictor by a suitable algorithm for the bandit setting like EXP4 (which considers a single hypothesis in each time-step). Similarly, if we want to capture inputs partitioned into several intervals,
> each resembling different hypothesis, we can use, for example, the classical SHARE algorithm
> instead. Furthermore, our analysis readily implies how the regret guarantees of these algorithms (e.g. EXP4 or SHARE) translate to approximation guarantees of the implied learning-augmented online algorithm.
>
> While in some scenarios it might be the case that the black-box predictor satisfies the two properties pointed out by the reviewer, in other scenarios it might not. Our framework allows to optimize and analyze the predictor explicitly towards achieving these properties and other desired properties.
>
> > There are no experiments, even simple synthetic ones. It would be interesting to see even in synthetic case if the framework can actually be carried out.
>
> Our work is proposing a new theoretical framework for modeling of learning augmented algorithms. We have decided to focus on theoretical analysis. Experimenting with our framework is an interesting direction for future research.
>
> > There are no algorithms in the main body and it is hard to judge algorithmic improvements.
>
> We have tried to informally convey the main ideas behind our algorithms in the main body. However we will try to further clarify these in subsequent versions (and welcome any suggestions toward this end).

---

### Decision · Program_Chairs · 2024-09-25

**Decision:**

Accept (poster)

**Comment:**

This paper considers the problem of online learning in the learning augmented model, focusing on the problems of caching, load-balancing, and non-clairvoyant scheduling. The authors propose a model where the learner has access to an underlying hypothesis class that it can use to augment it's prediction. The authors provide algorithms (often following existing algorithmic frameworks such as exponential weights) that exploit these predictions and give improved results compared to existing works considering such augmentations.

The reviewers felt this work provided a new approach to LA algorithms. There were some common concerns about the structure of the paper - eg, the fact that no algorithms are included in the body of the work. I share this concern, and I think at least one full concrete example (say for caching) should be included in the main body.